# Greedy and Random Quasi-Newton Methods with Faster Explicit Superlinear Convergence

**Dachao Lin**[1*]     **Haishan Ye**[2*]     **Zhihua Zhang**[3]

[1]Academy for Advanced Interdisciplinary Studies, Peking University
[2]School of Management, Xi'an Jiaotong University; Shenzhen Research Institute of Big Data
[3]School of Mathematical Sciences, Peking University; Huawei Noah's Ark Lab
`lindachao@pku.edu.cn`  `yehaishan@xjtu.edu.cn`  `zhzhang@math.pku.edu.cn.`

## Abstract

In this paper, we follow Rodomanov and Nesterov [19]'s work to study quasi-Newton methods. We focus on the common SR1 and BFGS quasi-Newton methods to establish better explicit (local) superlinear convergence rates. First, based on the *greedy quasi-Newton update* which greedily selects the direction to maximize a certain measure of progress, we improve the convergence rate to a condition-number-free superlinear convergence rate. Second, based on the *random quasi-Newton update* that selects the direction randomly from a spherically symmetric distribution, we show the same superlinear convergence rate established as above. Our analysis is closely related to the approximation of a given Hessian matrix, unconstrained quadratic objective, as well as the general strongly convex, smooth and strongly self-concordant functions.

## 1   Introduction

We study the superlinear convergence of famous quasi-Newton methods that replace the exact Hessian applied in classical Newton methods with certain approximations. The approximation is updated in iterations based on some special formulas from the previous variation. There exist various quasi-Newton algorithms with different Hessian approximations. The three most popular versions are the *Davidon-Fletcher-Powell (DFP) method* [7, 10], the *Broyden-Fletcher-Goldfarb-Shanno (BFGS) method* [2, 3, 9, 11, 21], and the *Symmetric Rank 1 (SR1) method* [1, 7], all of which belong to the Broyden family [1] of quasi-Newton algorithms. The most attractive property of quasi-Newton methods is their superlinear convergence, which can trace back to the 1970s [4, 8, 17]. However, the superlinear convergence rates provided in prior work are asymptotic [5, 13, 15, 22, 24]. The results only show that the ratio of successive residuals tends to zero as the running iterations approach to infinity, while still lacking a specific superlinear convergence rate.

Recently, Rodomanov and Nesterov [19] gave the first explicit local superlinear convergence for their proposed new quasi-Newton methods. They introduced greedy quasi-Newton updates by greedily selecting from basis vectors to maximize a certain measure of progress, and established an explicit non-asymptotic bound on the local superlinear convergence rate correspondingly. Their proofs are mainly applicable to the DFP methods because they reduced all possible Broyden family to the DFP update based on the monotonicity property (see Lemma 2.2). However, the SR1 and BFGS updates are more popular and faster than the DFP update in practice, which also has been verified in their experiments. In addition, Rodomanov and Nesterov [19] also discovered that the randomized methods do not lose superlinear convergence but they did not provide theoretical guarantees.

---

[*]Equal Contribution.

35th Conference on Neural Information Processing Systems (NeurIPS 2021).

| Greedy or Random Methods | Strongly Self-concordant Objective | $k_0$ |
|---|---|---|
| DFP, BFGS, SR1 [19] | $\left(1 - \frac{1}{n\kappa}\right)^{k(k-1)/2} \left(\frac{1}{2}\right)^k \left(1 - \frac{1}{2\kappa}\right)^{k_0}$ | $O\left(n\kappa \ln(n\kappa)\right)$ |
| BFGS, SR1 (Corollary 4.4) | $\left(1 - \frac{1}{n}\right)^{k(k-1)/2} \left(\frac{1}{2}\right)^k \left(1 - \frac{1}{2\kappa}\right)^{k_0}$ | $O\left(\max\{n, \kappa\} \ln(n\kappa)\right)$ |

Table 1: Comparison of the existing specific superlinear convergence rates of the random or greedy quasi-Newton methods with our results in the view of $\lambda_f(\cdot)$ (see the definition in Eq. (17)), where $n$ is the dimension of parameters, $\kappa$ is the condition number of the objective function, and $k_0, k$ are the iteration numbers. Rodomanov and Nesterov [19]'s results are suitable for greedy quasi-Newton methods using the Broyden family update, which includes DFP, BFGS, SR1 methods.

Along this line, there are other results of local superlinear convergence analysis. Rodomanov and Nesterov [20] analyzed the well-known DFP and BFGS methods, using a standard Hessian update direction through the previous variation. They demonstrated faster initial convergence rates, while slower final rates compared to Rodomanov and Nesterov [19]'s results. Rodomanov and Nesterov [18] improved Rodomanov and Nesterov [20]'s results by reducing the dependence of the condition number $\kappa$ to $\ln \kappa$, though having similar worse long-history behavior. Jin and Mokhtari [14] provided a non-asymptotic dimension-free superlinear convergence rate of the original Broyden family when the initial Hessian approximation is also good enough.

In this work, we prove faster convergence rates for greedy and random quasi-Newton algorithms, particularly on the SR1 and BFGS methods. We present our contribution as follows:

- First, in the setting of approximating an exact Hessian, we use two update methods: 1) *greedy update* modified from Rodomanov and Nesterov [19]; 2) *random update* that randomly selects the updating direction from any spherically symmetric distribution. We show that both methods share a faster condition-number-free convergence. Particularly, we obtain the superlinear convergence rate $O\left(1 - \frac{k}{n}\right)$ for SR1 update, and the linear convergence rate $O((1 - \frac{1}{n})^k)$ for BFGS update, where $k$ is the current iteration, and $n$ is the dimension of parameters. Both findings improve the original convergence rate $O((1 - \frac{1}{n\kappa})^k)$ [19].

- Second, we extend our analysis to a practical scheme. We show (local) superlinear convergence under the SR1 update and BFGS update, when applied to unconstrained quadratic objective or strongly self-concordant functions. We list our results in Table 1 with the same formulation of [19]. Note that in general, the convergence goes through two periods. The first period lasts for $k_0$ iterations, and only has a linear convergence rate $O((1 - \frac{1}{2\kappa})^{k_0})$. The second period has a superlinear convergence rate $O((1 - \frac{1}{n})^{k(k-1)/2})$. Our revised bound takes fewer first-period iterations $k_0$ as well as a faster (condition-number-free) superlinear convergence rate in the second period compared to [19].

**Notation.** We denote vectors by lowercase bold letters (e.g., $\boldsymbol{u}, \boldsymbol{x}$), and matrices by capital bold letters (e.g., $\boldsymbol{W} = [w_{ij}]$). We use $\boldsymbol{e}_1, \ldots, \boldsymbol{e}_n$ for the $n$-dimensional coordinate directions. Let $\lambda_1(\boldsymbol{A}) \geq \cdots \geq \lambda_n(\boldsymbol{A})$ be the eigenvalues of a real symmetric matrix $\boldsymbol{A} \in \mathbb{R}^{n \times n}$. Moreover, $\| \cdot \|$ denotes the $\ell_2$-norm (standard Euclidean norm). For a given positive definite matrix $\boldsymbol{A}$ (i.e., $\boldsymbol{A} \succ 0$), we induce the following pair of conjugate Euclidean norms: $\|\boldsymbol{x}\|_{\boldsymbol{A}} \triangleq \sqrt{\boldsymbol{x}^\top \boldsymbol{A} \boldsymbol{x}}$, $\|\boldsymbol{x}\|_{\boldsymbol{A}}^* \triangleq \sqrt{\boldsymbol{x}^\top \boldsymbol{A}^{-1} \boldsymbol{x}}$. When $\boldsymbol{A} = \nabla^2 f(\boldsymbol{x}) \succ 0$, we prefer to use notation $\| \cdot \|_{\boldsymbol{x}}$ and $\| \cdot \|_{\boldsymbol{x}}^*$, provided that there is no ambiguity with the reference function $f$. We recall the rate of convergence used in this paper.

**Definition 1.1 (Linear/Superlinear convergence)** *Suppose a sequence $\{x_n\}$ converges to $0$ with*

$$\lim_{n \to \infty} \frac{|x_{n+1}|}{|x_n|} = q \in [0, 1).$$

*Now suppose another sequence $\{y_n\}$ converges to $y^*$ and satisfies that $|y_n - y^*| \leq |x_n|, \forall n \geq 0$. We say $\{y_n\}$ converges superlinearly if $q = 0$, linearly if $q \in (0, 1)$.*

**Organization.** In Section 2, we discuss quasi-Newton updating rules for approximating a positive definite Hessian matrix using the SR1 update (Subsection 2.1) and the BFGS update (Subsection

2.2). In Section 3, we analyze previous methods applied to the problem of minimizing a quadratic function. In Section 4, we show that similar results also hold in a more general setting of minimizing the strongly self-concordant function. We give some empirical results in Section 5. Finally, in Section 6, we present our conclusion.

## 2 Quasi-Newton updates

Before starting our theoretical results, we briefly review a class of quasi-Newton updating rules for approximating a positive definite matrix $\boldsymbol{A} \in \mathbb{R}^{n \times n}$. We follow the definition in [19], employing the following family of updates which describes the Broyden family [16, Section 6.3] of quasi-Newton updates, parameterized by a scalar $\tau \in \mathbb{R}$.

**Definition 2.1** *Let $\boldsymbol{A} \preceq \boldsymbol{G}$ (that is, $\boldsymbol{G} - \boldsymbol{A}$ is positive semi-definite). For any $\boldsymbol{u} \in \mathbb{R}^n$, if $\boldsymbol{G}\boldsymbol{u} = \boldsymbol{A}\boldsymbol{u}$, define $\mathrm{Broyd}_\tau(\boldsymbol{G}, \boldsymbol{A}, \boldsymbol{u}) \triangleq \boldsymbol{G}$. Otherwise, we define*

$$
\begin{aligned}
\mathrm{Broyd}_\tau(\boldsymbol{G}, \boldsymbol{A}, \boldsymbol{u}) \triangleq{}& \tau \left[ \boldsymbol{G} - \frac{\boldsymbol{A}\boldsymbol{u}\boldsymbol{u}^\top \boldsymbol{G} + \boldsymbol{G}\boldsymbol{u}\boldsymbol{u}^\top \boldsymbol{A}}{\boldsymbol{u}^\top \boldsymbol{A}\boldsymbol{u}} + \left( \frac{\boldsymbol{u}^\top \boldsymbol{G}\boldsymbol{u}}{\boldsymbol{u}^\top \boldsymbol{A}\boldsymbol{u}} + 1 \right) \frac{\boldsymbol{A}\boldsymbol{u}\boldsymbol{u}^\top \boldsymbol{A}}{\boldsymbol{u}^\top \boldsymbol{A}\boldsymbol{u}} \right] \\
& + (1 - \tau) \left[ \boldsymbol{G} - \frac{(\boldsymbol{G} - \boldsymbol{A})\boldsymbol{u}\boldsymbol{u}^\top (\boldsymbol{G} - \boldsymbol{A})}{\boldsymbol{u}^\top (\boldsymbol{G} - \boldsymbol{A})\boldsymbol{u}} \right].
\end{aligned}
\tag{1}
$$

For several choices of $\tau$, we can recover several well-known quasi-Newton methods.

For $\tau = 0$, Eq. (1) corresponds to the well-known SR1 update:

$$
\mathrm{SR1}(\boldsymbol{G}, \boldsymbol{A}, \boldsymbol{u}) \triangleq \boldsymbol{G} - \frac{(\boldsymbol{G} - \boldsymbol{A})\boldsymbol{u}\boldsymbol{u}^\top (\boldsymbol{G} - \boldsymbol{A})}{\boldsymbol{u}^\top (\boldsymbol{G} - \boldsymbol{A})\boldsymbol{u}}.
\tag{2}
$$

For $\tau_{\mathrm{BFGS}} \triangleq \frac{\boldsymbol{u}^\top \boldsymbol{A}\boldsymbol{u}}{\boldsymbol{u}^\top \boldsymbol{G}\boldsymbol{u}} \in [0, 1]$, we recover the famous BFGS update:

$$
\mathrm{BFGS}(\boldsymbol{G}, \boldsymbol{A}, \boldsymbol{u}) \triangleq \boldsymbol{G} - \frac{\boldsymbol{G}\boldsymbol{u}\boldsymbol{u}^\top \boldsymbol{G}}{\boldsymbol{u}^\top \boldsymbol{G}\boldsymbol{u}} + \frac{\boldsymbol{A}\boldsymbol{u}\boldsymbol{u}^\top \boldsymbol{A}}{\boldsymbol{u}^\top \boldsymbol{A}\boldsymbol{u}}.
\tag{3}
$$

For $\tau = 1$, it corresponds to the well-known DFP update:

$$
\mathrm{DFP}(\boldsymbol{G}, \boldsymbol{A}, \boldsymbol{u}) \triangleq \boldsymbol{G} - \frac{\boldsymbol{A}\boldsymbol{u}\boldsymbol{u}^\top \boldsymbol{G} + \boldsymbol{G}\boldsymbol{u}\boldsymbol{u}^\top \boldsymbol{A}}{\boldsymbol{u}^\top \boldsymbol{A}\boldsymbol{u}} + \left( \frac{\boldsymbol{u}^\top \boldsymbol{G}\boldsymbol{u}}{\boldsymbol{u}^\top \boldsymbol{A}\boldsymbol{u}} + 1 \right) \frac{\boldsymbol{A}\boldsymbol{u}\boldsymbol{u}^\top \boldsymbol{A}}{\boldsymbol{u}^\top \boldsymbol{A}\boldsymbol{u}}.
\tag{4}
$$

Additionally, the Broyden family has matrix monotonicity below, showing the relationship among these quasi-Newton methods.

**Lemma 2.2 (Rodomanov and Nesterov [19] Lemmas 2.1 and 2.2)** *If $\boldsymbol{A} \preceq \boldsymbol{G} \preceq \eta \boldsymbol{A}$ for some $\eta \geq 1$, then we have for any $\boldsymbol{u} \in \mathbb{R}^n$, and $\tau_1, \tau_2 \in \mathbb{R}$ with $\tau_1 \leq \tau_2$ that*

$$
\mathrm{Broyd}_{\tau_1}(\boldsymbol{G}, \boldsymbol{A}, \boldsymbol{u}) \preceq \mathrm{Broyd}_{\tau_2}(\boldsymbol{G}, \boldsymbol{A}, \boldsymbol{u}).
$$

*And for any $\tau \in [0, 1]$, we have $\boldsymbol{A} \preceq \mathrm{Broyd}_\tau(\boldsymbol{G}, \boldsymbol{A}, \boldsymbol{u}) \preceq \eta \boldsymbol{A}$.*

Hence, if $\boldsymbol{A} \preceq \boldsymbol{G} \preceq \eta \boldsymbol{A}$ for some $\eta \geq 1$, it follows from Lemma 2.2 that

$$
\boldsymbol{A} \preceq \mathrm{SR1}(\boldsymbol{G}, \boldsymbol{A}, \boldsymbol{u}) \preceq \mathrm{BFGS}(\boldsymbol{G}, \boldsymbol{A}, \boldsymbol{u}) \preceq \mathrm{DFP}(\boldsymbol{G}, \boldsymbol{A}, \boldsymbol{u}) \preceq \eta \boldsymbol{A}.
$$

Intuitively, the approximation produced by SR1 is better than that produced by BFGS. And both of them are better than that produced by DFP. However, Rodomanov and Nesterov [19] deduced the analysis by casting all updates described by Broyden family ($\tau \in [0, 1]$) into the slowest DFP update ($\tau = 1$). Moreover, SR1 and BFGS methods also have faster numerical performance in practice. Therefore, we consider there might have faster superlinear convergence rates of SR1 and BFGS methods.

Now we consider the fundamental matrix approximation problem below to obtain the convergence rate of the SR1 and BFGS updates.

## 2.1 Superlinear convergence for SR1 update

We first describe the SR1 update for approximating a fixed positive definite matrix $\boldsymbol{A} \in \mathbb{R}^{n \times n}$ and leave the proof in Appendix A. Let us now justify the efficiency of update Eq. (2) in ensuring convergence $\boldsymbol{G}$ to $\boldsymbol{A}$. For this, we introduce the following measure of progress:

$$\tau_{\boldsymbol{A}}(\boldsymbol{G}) \triangleq \operatorname{tr}(\boldsymbol{G} - \boldsymbol{A}). \tag{5}$$

Previous work [19] uses a different measure as follows

$$\sigma_{\boldsymbol{A}}(\boldsymbol{G}) \triangleq \langle \boldsymbol{A}^{-1}, \boldsymbol{G} \rangle - n = \operatorname{tr}\left[(\boldsymbol{G} - \boldsymbol{A})\,\boldsymbol{A}^{-1}\right], \tag{6}$$

which might be improper for analyzing the SR1 update. Thus we employ a more concise measure $\tau_{\boldsymbol{A}}(\cdot)$. However, we will reuse $\sigma_{\boldsymbol{A}}(\cdot)$ in the proof of BFGS update in Subsection 2.2.

According to $\tau_{\boldsymbol{A}}(\boldsymbol{G})$, one iteration update leads to

$$\tau_{\boldsymbol{A}}(\boldsymbol{G}_+) \overset{(5)}{=} \tau_{\boldsymbol{A}}(\boldsymbol{G}) - \frac{\boldsymbol{u}^\top (\boldsymbol{G} - \boldsymbol{A})^2 \boldsymbol{u}}{\boldsymbol{u}^\top (\boldsymbol{G} - \boldsymbol{A}) \boldsymbol{u}}, \quad \boldsymbol{G}_+ = \mathrm{SR1}(\boldsymbol{G}, \boldsymbol{A}, \boldsymbol{u}).$$

Note that we always have $\boldsymbol{G}_+ \succeq \boldsymbol{A}$ if $\boldsymbol{G} \succeq \boldsymbol{A}$ from Lemma 2.2. Moreover, the choice of the updating direction $\boldsymbol{u}$ directly influences the decrease in the measure $\tau_{\boldsymbol{A}}(\cdot)$. In the following, we show two efficient ways for selecting $\boldsymbol{u}$ introduced in Algorithm 1.

First, we use the *greedy method* introduced in [19], that greedily selects $\boldsymbol{u}$ from the basis vectors to obtain the largest decrease of $\tau_{\boldsymbol{A}}(\boldsymbol{G}_+) - \tau_{\boldsymbol{A}}(\boldsymbol{G})$: $\bar{\boldsymbol{u}}_{\boldsymbol{A}}(\boldsymbol{G}) \triangleq \arg\max_{\boldsymbol{u} \in \{\boldsymbol{e}_1, \dots, \boldsymbol{e}_n\}} \frac{\boldsymbol{u}^\top (\boldsymbol{G} - \boldsymbol{A})^2 \boldsymbol{u}}{\boldsymbol{u}^\top (\boldsymbol{G} - \boldsymbol{A}) \boldsymbol{u}}$. However, we may encounter numerical overflow due to divided by 0 if $\boldsymbol{u}^\top (\boldsymbol{G} - \boldsymbol{A}) \boldsymbol{u}$ is nearly 0. Noting that $\boldsymbol{G} \succeq \boldsymbol{A}$, we have $\sqrt{(\boldsymbol{u}^\top (\boldsymbol{G} - \boldsymbol{A})^2 \boldsymbol{u})(\boldsymbol{u}^\top \boldsymbol{u})} \geq \boldsymbol{u}^\top (\boldsymbol{G} - \boldsymbol{A}) \boldsymbol{u} \geq 0, \ \forall \boldsymbol{u} \in \mathbb{R}^n$ from Cauchy–Schwarz inequality. Thus we employ a safer adjustment:

$$(\text{Greedy SR1}) \quad \bar{\boldsymbol{u}}_{\boldsymbol{A}}(\boldsymbol{G}) \triangleq \underset{\boldsymbol{u} \in \{\boldsymbol{e}_1, \dots, \boldsymbol{e}_n\}}{\arg\max} \frac{\boldsymbol{u}^\top (\boldsymbol{G} - \boldsymbol{A}) \boldsymbol{u}}{\boldsymbol{u}^\top \boldsymbol{u}} = \underset{\boldsymbol{u} \in \{\boldsymbol{e}_1, \dots, \boldsymbol{e}_n\}}{\arg\max} \boldsymbol{u}^\top (\boldsymbol{G} - \boldsymbol{A}) \boldsymbol{u}. \tag{7}$$

Moreover, we only need to obtain the diagonal elements of $\boldsymbol{A}$ (the current Hessian in practice), thus the total complexity is $O(n^2)$ in each iteration[2], which is acceptable and the same as the classical quasi-Newton methods.

Second, from the proof of the *greedy method*, we discover that the *random method* by choosing $\boldsymbol{u}$ from a spherically symmetric distribution, e.g.,

$$(\text{Random SR1}) \quad \boldsymbol{u} \sim \mathcal{N}(0, I_n) \text{ or } \boldsymbol{u} \sim \mathrm{Unif}(\mathcal{S}^{n-1}), \tag{8}$$

also has similar performance (in expectation) and the same running complexity $O(n^2)$ in each iteration.

Rodomanov and Nesterov [19, Theorem 3.5] already showed that $\boldsymbol{G}_n = \boldsymbol{A}$ if the $\boldsymbol{u}_k$ for $k \in \{0, \dots, n-1\}$ are linearly independent. Thus, we only consider $k \leq n$. Now let us estimate the decrease in the measure $\tau_{\boldsymbol{A}}(\cdot)$ based on the direction $\boldsymbol{u}$ chosen from previous strategies. In the following, the expectation considers all the randomness of the directions $\boldsymbol{u}_k$'s during iterations, and when applied to the *greedy method*, we can view it with no randomness for the same notation.

**Theorem 2.3** *Under the SR1 update in Algorithm 1, we obtain that for the greedy method defined in Eq. (7) or the random method defined in Eq. (8),*

$$0 \leq \mathbb{E}\,\tau_{\boldsymbol{A}}(\boldsymbol{G}_k) \leq \left(1 - \frac{1}{n - k + 1}\right) \mathbb{E}\,\tau_{\boldsymbol{A}}(\boldsymbol{G}_{k-1}) \leq \left(1 - \frac{k}{n}\right) \tau_{\boldsymbol{A}}(\boldsymbol{G}_0), \ 1 \leq k \leq n. \tag{9}$$

*Hence, $\tau_{\boldsymbol{A}}(\boldsymbol{G}_k)$ (or $\mathbb{E}\,\tau_{\boldsymbol{A}}(\boldsymbol{G}_k)$) converges to zero superlinearly.*

## 2.2 Linear convergence for BFGS update

We now consider the classical BFGS update in the same scheme and leave the proof in Appendix A. Using the measure $\sigma_{\boldsymbol{A}}(\cdot)$, we obtain that

$$\sigma_{\boldsymbol{A}}(\boldsymbol{G}_+) = \sigma_{\boldsymbol{A}}(\boldsymbol{G}) - \frac{\boldsymbol{u}^\top \boldsymbol{G} \boldsymbol{A}^{-1} \boldsymbol{G} \boldsymbol{u}}{\boldsymbol{u}^\top \boldsymbol{G} \boldsymbol{u}} + 1, \ \boldsymbol{G}_+ = \mathrm{BFGS}(\boldsymbol{G}, \boldsymbol{A}, \boldsymbol{u}). \tag{10}$$

---

[2]Note that we can use Hessian-vector product to obtain $\boldsymbol{A}\boldsymbol{u}$ in practice.

| **Algorithm 1** Greedy/Random SR1 update | **Algorithm 2** Greedy/Random BFGS update |
|---|---|
| 1: Initialization: Set $\boldsymbol{G}_0 \succeq \boldsymbol{A}$. | 1: Initialization: Set $\boldsymbol{G}_0 \succeq \boldsymbol{A}$, $\boldsymbol{L}_0 = \boldsymbol{G}_0^{-1/2}$. |
| 2: **for** $k = 0, \ldots, n-1$ **do** | 2: **for** $k \geq 0$ **do** |
| 3:    Choose $\boldsymbol{u}_k$ from | 3:    Compute $\boldsymbol{u}_k = \boldsymbol{L}_k^\top \tilde{\boldsymbol{u}}_k$ with $\tilde{\boldsymbol{u}}_k$ from |
|     1) *greedy method*: $\boldsymbol{u}_k = \bar{\boldsymbol{u}}_{\boldsymbol{A}}(\boldsymbol{G}_k)$, or |     1) *greedy method*: $\tilde{\boldsymbol{u}}_k = \tilde{\boldsymbol{u}}_{\boldsymbol{A}}(\boldsymbol{L}_k)$, or |
|     2) *random method*: $\boldsymbol{u}_k \sim \text{Unif}(\mathcal{S}^{n-1})$. |     2) *random method*: $\tilde{\boldsymbol{u}}_k \sim \text{Unif}(\mathcal{S}^{n-1})$. |
| 4:    Compute $\boldsymbol{G}_{k+1} = \text{SR1}(\boldsymbol{G}_k, \boldsymbol{A}, \boldsymbol{u}_k)$. | 4:    Compute $\boldsymbol{G}_{k+1} = \text{BFGS}(\boldsymbol{G}_k, \boldsymbol{A}, \boldsymbol{u}_k)$. |
| | 5:    Compute $\boldsymbol{L}_{k+1}$ based on Proposition 2.6. |
| 5: **end for** | 6: **end for** |

If we directly apply the *greedy method* or *random method* from the previous content, we could only obtain the same linear convergence rate in [19, Theorem 2.5]. However, if we take advantage of the current $\boldsymbol{G}$, and choose a scaled direction such that $\boldsymbol{u} = \boldsymbol{L}^\top \tilde{\boldsymbol{u}}$ where $\boldsymbol{L}^\top \boldsymbol{L} = \boldsymbol{G}^{-1}$, we could simplify the formulation and obtain a faster condition-number-free linear convergence rate. Specifically, the greedy update after replacing $\boldsymbol{u}$ to $\boldsymbol{L}^\top \tilde{\boldsymbol{u}}$ is as follows:

$$\text{(Greedy BFGS)} \ \ \tilde{\boldsymbol{u}}_{\boldsymbol{A}}(\boldsymbol{L}) = \underset{\tilde{\boldsymbol{u}} \in \{\boldsymbol{e}_1, \ldots, \boldsymbol{e}_n\}}{\arg\max} \frac{\tilde{\boldsymbol{u}}^\top \boldsymbol{L}^{-\top} \boldsymbol{A}^{-1} \boldsymbol{L}^{-1} \tilde{\boldsymbol{u}}}{\tilde{\boldsymbol{u}}^\top \tilde{\boldsymbol{u}}}, \tag{11}$$

and the random method is the same as the SR1 update used in Eq. (8).

$$\text{(Random BFGS)} \ \ \tilde{\boldsymbol{u}} \sim \mathcal{N}(0, I_n) \text{ or } \tilde{\boldsymbol{u}} \sim \text{Unif}(\mathcal{S}^{n-1}), \tag{12}$$

Now we give the linear convergence rate of our modified BFGS update.

**Theorem 2.4** *Under the BFGS update in Algorithm 2, we can obtain that for the greedy method defined in Eq.* (11) *or the random method defined in Eq.* (12),

$$0 \leq \mathbb{E}\, \sigma_A(\boldsymbol{G}_k) \leq \left(1 - \frac{1}{n}\right) \mathbb{E}\, \sigma_A(\boldsymbol{G}_{k-1}) \leq \left(1 - \frac{1}{n}\right)^k \sigma_A(\boldsymbol{G}_0), \ 1 \leq k. \tag{13}$$

*Therefore, $\sigma_A(\boldsymbol{G}_k)$ (or $\mathbb{E}\, \sigma_A(\boldsymbol{G}_k)$) converges to zero linearly.*

**Remark 2.5** *Note that the complexity in Eq.* (11) *is $O(n^3)$ because we have multiply-add operations with (unknown) $\boldsymbol{A}^{-1}$. Hence we **do not** apply this greedy strategy in practice, but view it as a theoretical result similar to the random strategy. Moreover, the random method is still practical, and we show the efficiency of our scaled direction compared to the original direction in experiments.*

Finally, we can employ an efficient way (with complexity $O(n^2)$) for updating $\boldsymbol{L}_k$. The main idea is employing the rank-one update and the one-row appending update of QR decomposition with the inverse BFGS update. In Proposition 2.6, we show how to compute $\boldsymbol{L}_{k+1}$ from $\boldsymbol{L}_k$ with $O(n^2)$ flops.

**Proposition 2.6** *Suppose we already have $\boldsymbol{H}_k \triangleq \boldsymbol{G}_k^{-1} = \boldsymbol{L}_k^\top \boldsymbol{L}_k$, where $\boldsymbol{L}_k$ is an upper triangular matrix. Now we construct $\boldsymbol{L}_{k+1}$ with $O(n^2)$ flops.*

*Step 1: Using the formulation in Golub and Van Loan [12, Section 12.5.1], we can obtain $QR$ decomposition of*

$$\boldsymbol{L}_k \left(\boldsymbol{I}_k - \frac{\boldsymbol{A}\boldsymbol{u}_k \boldsymbol{u}_k^\top}{\boldsymbol{u}_k^\top \boldsymbol{A}\boldsymbol{u}_k}\right) = \boldsymbol{L}_k - \frac{\boldsymbol{L}_k(\boldsymbol{A}\boldsymbol{u}_k)}{\boldsymbol{u}_k^\top \boldsymbol{A}\boldsymbol{u}_k} \boldsymbol{u}_k^\top$$

*with $O(n^2)$ flops because it is a rank-one change of $\boldsymbol{L}_k$.*

*Step 2: We have $\boldsymbol{L}_k \left(\boldsymbol{I}_n - \dfrac{\boldsymbol{A}\boldsymbol{u}_k \boldsymbol{u}_k^\top}{\boldsymbol{u}_k^\top \boldsymbol{A}\boldsymbol{u}_k}\right) = \boldsymbol{Q}_k \boldsymbol{R}_k$ from Step 1, with an orthogonal matrix $\boldsymbol{Q}_k \in \mathbb{R}^{n \times n}$ and an upper triangular matrix $\boldsymbol{R}_k \in \mathbb{R}^{n \times n}$. Denoting $\boldsymbol{v}_k = \dfrac{\boldsymbol{u}_k}{\sqrt{\boldsymbol{u}_k^\top \boldsymbol{A}\boldsymbol{u}_k}}$, we can write*

$$\boldsymbol{H}_{k+1} \overset{(16)}{=} \boldsymbol{R}_k^\top \boldsymbol{Q}_k^\top \boldsymbol{Q}_k \boldsymbol{R}_k + \frac{\boldsymbol{u}_k \boldsymbol{u}_k^\top}{\boldsymbol{u}_k^\top \boldsymbol{A}\boldsymbol{u}_k} = \boldsymbol{R}_k^\top \boldsymbol{R}_k + \boldsymbol{v}_k \boldsymbol{v}_k^\top = \begin{pmatrix} \boldsymbol{v}_k & \boldsymbol{R}_k^\top \end{pmatrix} \begin{pmatrix} \boldsymbol{v}_k^\top \\ \boldsymbol{R}_k \end{pmatrix}.$$

*Using the formulation in Golub and Van Loan [12, Section 12.5.3] for computing the $QR$ decomposition after appending a row, we can obtain $\begin{pmatrix} \boldsymbol{v}_k^\top \\ \boldsymbol{R}_k \end{pmatrix} = \boldsymbol{Q}_{k+1} \boldsymbol{R}_{k+1}$ with only $O(n^2)$ flops, where $\boldsymbol{Q}_{k+1} \in \mathbb{R}^{(n+1) \times n}$ is a column orthogonal matrix and $\boldsymbol{R}_{k+1} \in \mathbb{R}^{n \times n}$ is an upper triangular matrix. This implies $\boldsymbol{H}_{k+1} = \boldsymbol{R}_{k+1}^\top \boldsymbol{R}_{k+1}$, which satisfies our requirements.*

---

**Algorithm 3** Greedy/Random SR1/BFGS methods for quadratic minimization

---
1: Initialization: Choose $\boldsymbol{x}_0 \in \mathbb{R}^n$. Set $\boldsymbol{G}_0 = L\boldsymbol{I}_n$ (or any $\boldsymbol{G}_0 \succeq \boldsymbol{A}$).
2: **for** $k \geq 0$ **do**
3:     Update $\boldsymbol{x}_{k+1} = \boldsymbol{x}_k - \boldsymbol{G}_k^{-1}\nabla f(\boldsymbol{x}_k)$. Choose one of the following update rules:
4:     (i) SR1: Choose $\boldsymbol{u}_k$ following Algorithm 1. Compute $\boldsymbol{G}_{k+1} = \mathrm{SR1}(\boldsymbol{G}_k, \boldsymbol{A}, \boldsymbol{u}_k)$.
5:     (ii) BFGS: Choose $\boldsymbol{u}_k$ following Algorithm 2. Compute $\boldsymbol{G}_{k+1} = \mathrm{BFGS}(\boldsymbol{G}_k, \boldsymbol{A}, \boldsymbol{u}_k)$.
6: **end for**

---

## 3 Unconstrained quadratic minimization

Based on the efficiency of the greedy/random SR1 and BFGS updates in matrix approximation, we next turn to minimize the quadratic function (with a fixed Hessian):

$$f(\boldsymbol{x}) = \frac{1}{2}\boldsymbol{x}^\top \boldsymbol{A}\boldsymbol{x} - \boldsymbol{b}^\top \boldsymbol{x}, \tag{14}$$

where $\boldsymbol{A}$ is a positive definite matrix, and there exist $\mu, L > 0$, s.t., $\mu\boldsymbol{I}_n \preceq \boldsymbol{A} \preceq L\boldsymbol{I}_n$. We show the detail in Algorithm 3, which is only for theoretical analysis. In practice, we use the inverse update rules (Nocedal and Wright [16, Eq. (6.17) and Eq. (6.25)]) to update $\boldsymbol{G}_k^{-1}$ directly:

$$\boldsymbol{G}_+^{-1} = \boldsymbol{G}^{-1} + \frac{(\boldsymbol{I}_n - \boldsymbol{G}^{-1}\boldsymbol{A})\boldsymbol{u}\boldsymbol{u}^\top(\boldsymbol{I}_n - \boldsymbol{A}\boldsymbol{G}^{-1})}{\boldsymbol{u}^\top(\boldsymbol{A} - \boldsymbol{A}\boldsymbol{G}^{-1}\boldsymbol{A})\boldsymbol{u}}, \quad \boldsymbol{G}_+ = \mathrm{SR1}(\boldsymbol{G}, \boldsymbol{A}, \boldsymbol{u}); \tag{15}$$

$$\boldsymbol{G}_+^{-1} = \left(\boldsymbol{I}_n - \frac{\boldsymbol{u}\boldsymbol{u}^\top\boldsymbol{A}}{\boldsymbol{u}^\top\boldsymbol{A}\boldsymbol{u}}\right)\boldsymbol{G}^{-1}\left(\boldsymbol{I}_n - \frac{\boldsymbol{A}\boldsymbol{u}\boldsymbol{u}^\top}{\boldsymbol{u}^\top\boldsymbol{A}\boldsymbol{u}}\right) + \frac{\boldsymbol{u}\boldsymbol{u}^\top}{\boldsymbol{u}^\top\boldsymbol{A}\boldsymbol{u}}, \quad \boldsymbol{G}_+ = \mathrm{BFGS}(\boldsymbol{G}, \boldsymbol{A}, \boldsymbol{u}). \tag{16}$$

To estimate the convergence rate of Scheme (14), we measure the norm of the gradient of $f$ as

$$\lambda_f(\boldsymbol{x}) \triangleq \sqrt{\nabla f(\boldsymbol{x})^\top \nabla^2 f(\boldsymbol{x})^{-1}\nabla f(\boldsymbol{x})}, \ \boldsymbol{x} \in \mathbb{R}^n. \tag{17}$$

Note that this measure of optimality is directly related to the functional residual. Indeed, let $\boldsymbol{x}^* = \boldsymbol{A}^{-1}\boldsymbol{b}$ be the minimizer of Eq. (14). Then we obtain

$$f(\boldsymbol{x}) - f(\boldsymbol{x}^*) = \frac{1}{2}\left(\boldsymbol{x} - \boldsymbol{x}^*\right)^\top \boldsymbol{A}\left(\boldsymbol{x} - \boldsymbol{x}^*\right) = \frac{1}{2}\left(\boldsymbol{A}\boldsymbol{x} - \boldsymbol{b}\right)^\top \boldsymbol{A}^{-1}\left(\boldsymbol{A}\boldsymbol{x} - \boldsymbol{b}\right) = \frac{1}{2}\lambda_f(\boldsymbol{x})^2.$$

The following lemma shows how $\lambda_f(\cdot)$ varies after one iteration of process in Algorithm 3.

**Lemma 3.1 (Rodomanov and Nesterov [19] Lemma 3.2)** *Let $k \geq 0$, and let $\eta_k \geq 1$ satisfy $\boldsymbol{A} \preceq \boldsymbol{G}_k \preceq \eta_k\boldsymbol{A}$. Then we have that $\lambda_f(\boldsymbol{x}_{k+1}) \leq \left(1 - \frac{1}{\eta_k}\right)\lambda_f(\boldsymbol{x}_k) \leq (\eta_k - 1)\lambda_f(\boldsymbol{x}_k)$.*

Thus, to estimate how fast $\lambda_f(\boldsymbol{x}_k)$ converges to 0, we need the upper bound of $\eta_k$, which is already done in Subsections 2.1 and 2.2. Thus, we can guarantee a superlinear convergence of $\lambda_f(\boldsymbol{x}_k)$ using the greedy/random SR1 or BFGS update. The proof of Theorem 3.2 can be found in Appendix A.

**Theorem 3.2** *Under Algorithm 3, if we choose SR1 update, then we have*

$$\mathbb{E}\frac{\lambda_f(\boldsymbol{x}_{k+1})}{\lambda_f(\boldsymbol{x}_k)} \leq \left(1 - \frac{k}{n}\right)\frac{\mathrm{tr}(\boldsymbol{G}_0 - \boldsymbol{A})}{\mu}, \ 0 \leq k \leq n.$$

*If we adopt BFGS update, then we have*

$$\mathbb{E}\frac{\lambda_f(\boldsymbol{x}_{k+1})}{\lambda_f(\boldsymbol{x}_k)} \leq \left(1 - \frac{1}{n}\right)^k \mathrm{tr}\left[(\boldsymbol{G}_0 - \boldsymbol{A})\boldsymbol{A}^{-1}\right], \ k \geq 0.$$

Thus, we see both SR1 and BFGS methods share superlinear convergence for $\{\lambda_f(\boldsymbol{x}_k)\}$. In particular, for the SR1 update, our bound recovers the classical result in [16, Theorem 6.1], showing that the update stops after finite steps because $\boldsymbol{G}_n = \boldsymbol{A}$ and $\lambda_f(\boldsymbol{x}_{n+1}) = 0$. Moreover, we also give an exact convergence rate during the entire optimization process. And the main decreasing term $\left(1 - \frac{k}{n}\right)$ for the SR1 update as well as $(1 - \frac{1}{n})^k$ for the BFGS update in the $k$-th iteration are independent of the condition number $\kappa$ of $\boldsymbol{A}$, which improves the bound $(1 - \frac{1}{n\kappa})^k$ in [19, Theorem 3.4].

**Algorithm 4** Greedy/Random SR1/BFGS methods for strongly self-concordant objective

1: Initialization: Choose $\boldsymbol{x}_0 \in \mathbb{R}^n$. Set $\boldsymbol{G}_0 = L\boldsymbol{I}_n, \boldsymbol{L}_0 = \boldsymbol{I}_n/\sqrt{L}$.
2: **for** $k \geq 0$ **do**
3:     Update $\boldsymbol{x}_{k+1} = \boldsymbol{x}_k - \boldsymbol{G}_k^{-1}\nabla f(\boldsymbol{x}_k)$.
4:     Compute $r_k = \|\boldsymbol{x}_{k+1} - \boldsymbol{x}_k\|_{\boldsymbol{x}_k}$, $\tilde{\boldsymbol{G}}_k = (1 + Mr_k)\boldsymbol{G}_k$, $\tilde{\boldsymbol{L}}_k = \boldsymbol{L}_k/\sqrt{1 + Mr_k}$.
5:     (i) Greedy/Random SR1: Choose $\boldsymbol{u}_k = \bar{\boldsymbol{u}}_{\nabla^2 f(\boldsymbol{x}_{k+1})}(\tilde{\boldsymbol{G}}_k)$, or $\boldsymbol{u}_k \sim \mathrm{Unif}(\mathcal{S}^{n-1})$.
    Compute $\boldsymbol{G}_{k+1} = \mathrm{SR1}(\tilde{\boldsymbol{G}}_k, \nabla^2 f(\boldsymbol{x}_{k+1}), \boldsymbol{u}_k)$.
6:     (ii) Greedy/Random BFGS: Choose $\tilde{\boldsymbol{u}}_k = \tilde{\boldsymbol{u}}_{\nabla^2 f(\boldsymbol{x}_{k+1})}(\tilde{\boldsymbol{L}}_k)$, or $\tilde{\boldsymbol{u}}_k \sim \mathrm{Unif}(\mathcal{S}^{n-1})$.
    Compute $\boldsymbol{G}_{k+1} = \mathrm{BFGS}(\tilde{\boldsymbol{G}}_k, \nabla^2 f(\boldsymbol{x}_{k+1}), \tilde{\boldsymbol{L}}_k^\top \tilde{\boldsymbol{u}}_k)$, and $\boldsymbol{L}_{k+1}$ based on Proposition 2.6.
7: **end for**

## 4 Minimization of general functions

Finally, we consider the optimization of a more general machine learning objective with unfixed Hessians: $\min_{\boldsymbol{x}\in\mathbb{R}^n} f(\boldsymbol{x})$, where $f : \mathbb{R}^n \to \mathbb{R}$ is a twice differentiable function with positive definite Hessians. Our goal is to extend the results in the previous sections, assuming that the methods can start from a sufficiently good initial point $\boldsymbol{x}_0$. We use the same assumption *strongly self-concordant* followed by Rodomanov and Nesterov [19].

**Definition 4.1 (Strongly self-concordant)** *A function $f : \mathbb{R}^n \to \mathbb{R}$ is strongly self-concordant if the Hessians of $f$ are close to each other in the sense that there exists a constant $M \geq 0$ s.t.*

$$\nabla^2 f(\boldsymbol{y}) - \nabla^2 f(\boldsymbol{x}) \preceq M\|\boldsymbol{y} - \boldsymbol{x}\|_{\boldsymbol{z}}\nabla^2 f(\boldsymbol{w}), \ \forall \, \boldsymbol{x}, \boldsymbol{y}, \boldsymbol{z}, \boldsymbol{w} \in \mathbb{R}^n.$$

Rodomanov and Nesterov [19] have already mentioned several properties and examples of strongly self-concordant function, such as a strongly convex function with Lipschitz continuous Hessian. Let us now make one more common assumption about the function $f$ as Rodomanov and Nesterov [19] did, that the function $f$ is $\mu$-strongly convex and $L$-smooth, i.e., there exist $L \geq \mu > 0$ such that

$$\mu\boldsymbol{I}_n \preceq \nabla^2 f(\boldsymbol{x}) \preceq L\boldsymbol{I}_n, \ \forall \, \boldsymbol{x} \in \mathbb{R}^n$$

and $\kappa := L/\mu$. Unlike quadratic minimization, the true Hessian in each step varies. In order to make $\boldsymbol{G}_k \succeq \nabla^2 f(\boldsymbol{x}_k)$ always hold, we adjust $\boldsymbol{G}_k$ before doing quasi-Newton update. Instructed from [19], we also use the *correction strategy*, which enlarges the approximation $\boldsymbol{G}_k$ properly shown in Line 4 of Algorithm 4. Moreover, Algorithm 4 is only for theoretical analysis. We need to adopt the inverse update rules (Eqs. (15) and (16)) in practice. Additionally, we assume that the constants $M$ and $L$ are available for simplicity.

For the BFGS update, we can analyze how the Hessian approximation measure $\sigma_{\boldsymbol{x}}(\boldsymbol{G}) \triangleq \sigma_{\nabla^2 f(\boldsymbol{x})}(\boldsymbol{G})$ changes after one iteration following [19]. The proof can be found in Appendix A.

**Theorem 4.2** *Suppose that in Algorithm 4 BFGS update is used, and that the initial point $\boldsymbol{x}_0$ is sufficiently close to the solution:*

$$M\lambda_f(\boldsymbol{x}_0) \leq \frac{\ln 2}{4\kappa(2n+1)}. \tag{18}$$

*Then for all $k \geq 0$, we have*

$$\nabla^2 f(\boldsymbol{x}_k) \preceq \boldsymbol{G}_k, \ \mathbb{E}\,\sigma_{\boldsymbol{x}_k}(\boldsymbol{G}_k) \leq 2n\kappa\left(1 - \frac{1}{n}\right)^k, \tag{19}$$

*and*

$$\mathbb{E}\,\frac{\lambda_f(\boldsymbol{x}_{k+1})}{\lambda_f(\boldsymbol{x}_k)} \leq 2n\kappa\left(1 - \frac{1}{n}\right)^k. \tag{20}$$

As for the SR1 update, the main difficulty compared to [19] is that we utilize a different measure $\tau_{\boldsymbol{A}}(\cdot)$ instead of $\sigma_{\boldsymbol{A}}(\cdot)$, resulting in a refined proof version below, and we leave the proofs in Appendix A.

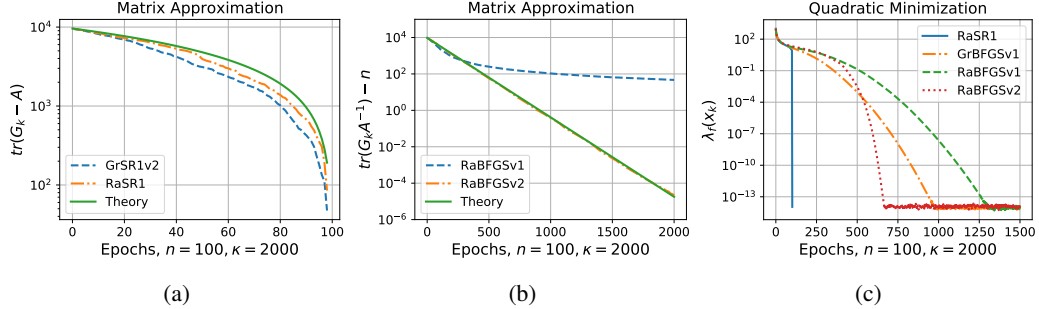

Figure 1: (a, b) Comparison of different direction choosing methods under the SR1 or BFGS update for approximating a matrix $\boldsymbol{A}$ that $\mu \boldsymbol{I}_n \preceq \boldsymbol{A} \preceq L\boldsymbol{I}_n$ from $\boldsymbol{G}_0 = L\boldsymbol{I}_n$. (a) The variation of $\tau_{\boldsymbol{A}}(\boldsymbol{G}_k)$ during Random SR1 (RaSR1) and our Greedy SR1 (GrSR1v2) update with nearly matched upper bound. (b) The variation of $\sigma_{\boldsymbol{A}}(\boldsymbol{G}_k)$ during our Random BFGS (RaBFGSv2) update and the original random version (RaBFGSv1). (c) Comparison of SR1 and BFGS methods for quadratic objective. Here we only depict RaSR1 method, while the other SR1-type methods share similar behavior.

**Theorem 4.3** *For Algorithm 4 with SR1 update, suppose that the initial point $\boldsymbol{x}_0$ is sufficiently close to the solution:*

$$M\lambda_f(\boldsymbol{x}_0) \leq \frac{\ln 2}{4\kappa(2n\kappa + 1)}. \tag{21}$$

*Then for all $k \geq 0$, we have*

$$\nabla^2 f(\boldsymbol{x}_k) \preceq \boldsymbol{G}_k, \mathbb{E}\,\sigma_{\boldsymbol{x}_k}(\boldsymbol{G}_k) \leq 2n\kappa^2\left(1 - \frac{1}{n}\right)^k, \tag{22}$$

*and*

$$\mathbb{E}\,\frac{\lambda_f(\boldsymbol{x}_{k+1})}{\lambda_f(\boldsymbol{x}_k)} \leq 2n\kappa^2\left(1 - \frac{1}{n}\right)^k. \tag{23}$$

**Corollary 4.4** *Combining with [19, Theorem 4.7] (shown in Appendix B.1), if $\boldsymbol{x}_0$ satisfies $M\lambda_f(\boldsymbol{x}_0) \leq \frac{\ln \frac{3}{2}}{4\kappa}$, we could obtain 1) for the greedy BFGS or greedy SR1 method,*

$$\lambda_f(\boldsymbol{x}_{k_0+k}) \leq \left(1 - \frac{1}{n}\right)^{k(k-1)/2} \cdot \left(\frac{1}{2}\right)^k \cdot \left(1 - \frac{1}{2\kappa}\right)^{k_0} \cdot \lambda_f(\boldsymbol{x}_0), \text{ for all } k \geq 0,$$

*where $k_0 = O\left(\max\{n, \kappa\}\ln(n\kappa)\right)$, 2) for the random BFGS or random SR1 method, with probability $1 - \delta$ for any $\delta \in (0, 1)$,*

$$\lambda_f(\boldsymbol{x}_{k_0+k}) \leq \left(1 - \frac{1}{n+1}\right)^{k(k-1)/2} \cdot \left(\frac{1}{2}\right)^k \cdot \left(1 - \frac{1}{2\kappa}\right)^{k_0} \cdot \lambda_f(\boldsymbol{x}_0), \text{ for all } k \geq 0,$$

*where $k_0 = O\left(\max\{n, \kappa\}\ln(n\kappa/\delta)\right)$.*

Therefore, both greedy and random methods have non-asymptotic superlinear convergence rates. Additionally, our superlinear rates are condition-number-free compared to the rates in [19].

## 5 Numerical experiments

In this section, we verify our theorems through numerical results for quasi-Newton methods. Rodomanov and Nesterov [19] have already compared their proposed greedy quasi-Newton method with the classical quasi-Newton methods. They showed that GrDFP, GrBFGS, GrSR1 (greedy DFP, BFGS, SR1 methods) using directions based on $\bar{\boldsymbol{u}}_{\boldsymbol{A}}(\boldsymbol{G}) = \arg\max_{\boldsymbol{u} \in \{\boldsymbol{e}_1, \ldots, \boldsymbol{e}_n\}} \frac{\boldsymbol{u}^\top \boldsymbol{G}\boldsymbol{u}}{\boldsymbol{u}^\top \boldsymbol{A}\boldsymbol{u}}$, have quite competitive convergence with the standard versions. They also presented the results for the randomized versions RaDFP, RaBFGS, RaSR1, which choose directions directly from a standard Euclidean sphere. They discovered that the randomized methods are slightly slower than the greedy ones. However, the difference is not really significant.

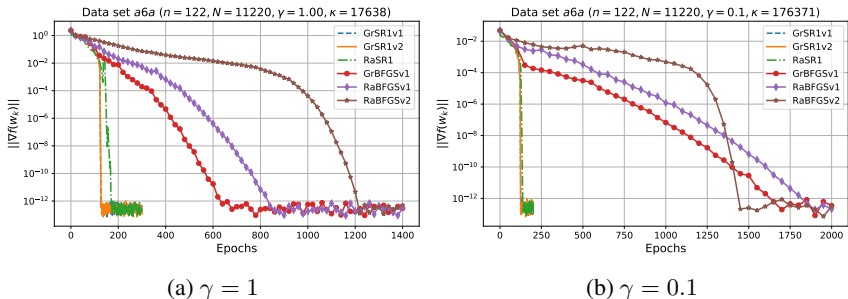

(a) $\gamma = 1$            (b) $\gamma = 0.1$

Figure 2: Comparison of SR1 and BFGS update for $\ell_2$-regularized logistic regression applied with 'a6a' data from the LIBSVM collection of real-world data sets. We list the name of dataset, the dimension $n$ and the condition number $\kappa$ under the corresponding $\gamma$ in the title of each figure. The lines of GrSR1v1 and GrSR1v2 are overlapped in some figures.

The difference between our algorithms and their methods mainly comes from the greedy strategy for SR1 and the random strategy for BFGS[3]. Hence, we mainly focus on exhibiting our validity in these schemes. We refer to GrSR1v2 as our revised method and GrSR1v1 as the previous method. Similarly, we denote RaBFGSv2 that uses scaled directions ($\boldsymbol{L}^\top \boldsymbol{u}$) and RaBFGSv1 that directly uses random directions $\boldsymbol{u}$ correspondingly. We use $\boldsymbol{u} \sim \mathrm{Unif}(\mathcal{S}^{n-1})$ in all experiments for brevity.

**Matrix approximation.** When using Algorithms 1 and 2 for approximating a matrix $\boldsymbol{A} \succ 0$, we show the measure as proved by Theorems 2.3 and 2.4 in Figures 1a and 1b. As Figure 1a depicts, our greedy and random SR1 updates (GrSR1v2 and RaSR1) share superlinear convergence rate under measure $\tau_{\boldsymbol{A}}(\cdot)$, while our theoretical bound matches them well. Moreover, Figure 1b describes the behavior of the random BFGS update. Note that the greedy BFGS update we proposed does not satisfy the $O(n^2)$ complexity, thus we leave it out. Our theory totally matches the linear convergence of measure $\sigma_{\boldsymbol{A}}(\cdot)$ in our modified random BFGS update (RaBFGSv2). However, directly choosing a direction without scaling (RaBFGSv1) fails to give such bounds. Moreover, we also discover the same findings under different condition numbers in Appendix C.1. Clearly, our methods provide effective ways of approaching a positive definite Hessian matrix.

**Quadratic minimization.** We also consider unconstrained quadratic minimization with the same positive definite matrix $\boldsymbol{A}$ and a randomly selected vector $\boldsymbol{b} \in \mathbb{R}^n$. Running Algorithm 3 with SR1 and BFGS updates, we obtain the superlinear convergence of $\lambda_f(\cdot)$ shown in Figure 1c. Not surprisingly, our RaBFGSv2 runs faster than RaBFGSv1, while we also have the theoretical guarantee. At the same time, SR1-type methods converge to zero after $n + 1$ steps because $\boldsymbol{G}_n = \boldsymbol{A}$. Here, we only depict the RaSR1 update, while the other SR1-type methods share similar behavior. Although our theoretical bound can not directly match the experiments due to the related initial terms $\tau_{\boldsymbol{A}}(\boldsymbol{G}_0)$ and $\sigma_{\boldsymbol{A}}(\boldsymbol{G}_0)$, the decay terms: $(1 - k/n)$ vs. $(1 - 1/n)^k$ already show the superiority of the SR1 method over the BFGS method in the quadratic minimization problem.

**Regularized logistic regression.** Next, we consider $\ell_2$-regularized logistic regression:

$$f(\boldsymbol{w}) = \sum_{i=1}^{N} \ln \left( 1 + e^{-y_i \boldsymbol{w}^\top \boldsymbol{x}_i} \right) + \frac{\gamma}{2} \|\boldsymbol{w}\|^2, \ \boldsymbol{w} \in \mathbb{R}^n,$$

where $\boldsymbol{X} = [\boldsymbol{x}_1, \ldots, \boldsymbol{x}_N] \in \mathbb{R}^{n \times N}$ are training samples with the corresponding labels $y_1, \ldots, y_N \in \{+1, -1\}$, and $\gamma > 0$ is the regularization coefficient. We follow the same experimental design but take data from the LIBSVM collection of real-world data sets for binary classification problems [6]. And we do not apply the *correction strategy* ($\tilde{\boldsymbol{G}}_k = (1 + M r_k) \boldsymbol{G}_k$ in Algorithm 4) recommended by [19]. Other details are shown in Appendix C.2. In order to simulate the local convergence, we use the same initialization after running several standard Newton's steps to make measure $\|\nabla f(\boldsymbol{w}_0)\|$ small (around $10^{-2} \sim 10^1$).

---

[3]There is no difference in the random SR1 method compared to Rodomanov and Nesterov [19], which directly selects random directions. And our greedy BFGS method is not efficient ($O(n^3)$ in each iteration) as we mentioned in Remark 2.5. Thus we leave it out.

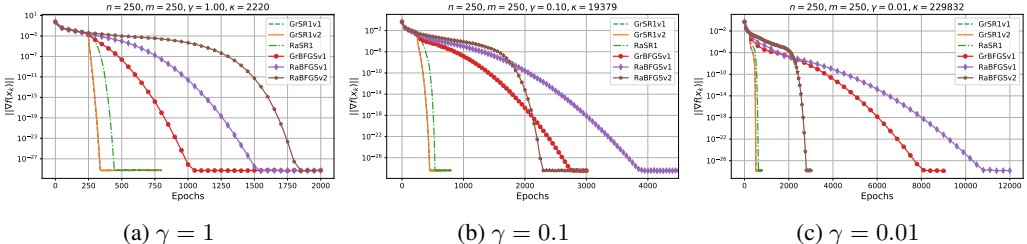

(a) $\gamma = 1$      (b) $\gamma = 0.1$      (c) $\gamma = 0.01$

Figure 3: Comparison of SR1 and BFGS update for Regularized Log-Sum-Exp. The dimension $n$, the number $m$ of linear functions, the regularization coefficient $\gamma$ and condition number $\kappa$ are displayed in the title of each graph. The lines of GrSR1v1 and GrSR1v2 are overlapped in each figure.

We show the results in Figure 2. As we can see, the general picture is the same as for the previous objective. In particular, SR1-type methods are faster than BFGS-type methods, and the greedy algorithms also converge more rapidly than the random algorithms. The only difference is that our RaBFGSv2 may have slower convergence behavior than RaBFGSv1 under a small $\kappa$ in Figure 2a. We consider our scaled direction is more suitable for a constant Hessian matrix as the quadratic objective has. Thus we still have a better convergence rate in the last few iterations when Hessians are nearly unchanged in Figure 2a. However, the Hessian varies drastically in the initial period. Thus there is less benefit under a more accurate Hessian approximation. When applied to the ill-conditioning setting with a large $\kappa$ in Figure 2b, we find our RaBFGSv2 could be faster than GrBFGSv1 and RaBFGSv1. This implies that our proposed method has less dependence on the condition number $\kappa$.

**Regularized Log-Sum-Exp.**  Followed by Rodomanov and Nesterov [19], we also present preliminary computational results for greedy and random quasi-Newton methods, applied to the following test function with $\boldsymbol{C} = [\boldsymbol{c}_1, \ldots, \boldsymbol{c}_m] \in \mathbb{R}^{n \times m}$, $b_1, \ldots, b_m \in \mathbb{R}$, and $\gamma > 0$:

$$f(\boldsymbol{x}) := \ln \left( \sum_{j=1}^{m} e^{\boldsymbol{c}_j^\top \boldsymbol{x} - b_j} \right) + \frac{1}{2} \sum_{j=1}^{m} \left( \boldsymbol{c}_j^\top \boldsymbol{x} \right)^2 + \frac{\gamma}{2} \|\boldsymbol{x}\|^2, \boldsymbol{x} \in \mathbb{R}^n.$$

We also use the same synthetic data in [19, Section 5.1] and leave the detail in Appendix C.2. As Figure 3 depicts, the BFGS-type methods are slower than the SR1-type methods. GrBFGSv1 and RaBFGSv1 are faster than RaBFGSv2 in a small condition number case, but they become slower than RaBFGSv2 when the condition number becomes huge. Therefore, we think our RaBFGSv2 which uses scaled direction indeed has less dependence on the condition number as our theory shows.

Overall, our proposed methods do not lose the superlinear convergence rate in the large condition number schemes, while we also present the theoretical guarantee for these algorithms.

## 6 Conclusion

In this work, we have studied the behavior of two famous quasi-Newton methods: the SR1 and the BFGS methods. We have presented different greedy methods in contrast to Rodomanov and Nesterov [19], and the random version of these methods. In particular, we have provided the faster Hessian approximation behavior and the condition-number-free (local) superlinear convergence rates applied to quadratic or strongly self-concordant objectives. Moreover, the experiments corroborate our analysis well. Note that our current results do not give the superlinear convergence rate of the standard quasi-Newton methods with only gradient information. However, we hope that the theoretical analysis and the related work would be useful for understanding the quasi-Newton methods in a more specific view.

## Acknowledgments and Disclosure of Funding

Lin and Zhang have been supported by the National Key Research and Development Project of China (No. 2018AAA0101004), by the National Natural Science Foundation of China (No. 11771002), and Beijing Natural Science Foundation (Z190001). Ye has been supported by the National Natural Science Foundation of China (No. 12101491), and Shenzhen Research Institute of Big Data (SRIBD) (No. J00120190004).

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
