$u_k$ from | 3:    Compute $u_k = L_k^\top \tilde{u}_k$ with $\tilde{u}_k$ from |
|     1) *greedy method*: $u_k = \bar{u}_A(G_k)$, or |     1) *greedy method*: $\tilde{u}_k = \tilde{u}_A(L_k)$, or |
|     2) *random method*: $u_k \sim \mathrm{Unif}(\mathcal{S}^{n-1})$. |     2) *random method*: $\tilde{u}_k \sim \mathrm{Unif}(\mathcal{S}^{n-1})$. |
| 4:    Compute $G_{k+1} = \mathrm{SR1}(G_k, A, u_k)$. | 4:    Compute $G_{k+1} = \mathrm{BFGS}(G_k, A, u_k)$. |
|  | 5:    Compute $L_{k+1}$ based on Proposition 2.6. |
| 5: **end for** | 6: **end for** |

If we directly apply the *greedy method* or *random method* from the previous content, we could only obtain the same linear convergence rate in [19, Theorem 2.5]. However, if we take advantage of the current $G$, and choose a scaled direction such that $u = L^\top \tilde{u}$ where $L^\top L = G^{-1}$, we could simplify the formulation and obtain a faster condition-number-free linear convergence rate. Specifically, the greedy update after replacing $u$ to $L^\top \tilde{u}$ is as follows:

$$\text{(Greedy BFGS)} \ \ \tilde{u}_A(L) = \underset{\tilde{u} \in \{e_1, \ldots, e_n\}}{\arg\max} \frac{\tilde{u}^\top L^{-\top} A^{-1} L^{-1} \tilde{u}}{\tilde{u}^\top \tilde{u}}, \tag{11}$$

and the random method is the same as the SR1 update used in Eq. (8).

$$\text{(Random BFGS)} \ \ \tilde{u} \sim \mathcal{N}(0, I_n) \text{ or } \tilde{u} \sim \mathrm{Unif}(\mathcal{S}^{n-1}), \tag{12}$$

Now we give the linear convergence rate of our modified BFGS update.

**Theorem 2.4** *Under the BFGS update in Algorithm 2, we can obtain that for the greedy method defined in Eq.* (11) *or the random method defined in Eq.* (12),

$$0 \leq \mathbb{E}\, \sigma_A(G_k) \leq \left(1 - \frac{1}{n}\right) \mathbb{E}\, \sigma_A(G_{k-1}) \leq \left(1 - \frac{1}{n}\right)^k \sigma_A(G_0), \ 1 \leq k. \tag{13}$$

*Therefore,* $\sigma_A(G_k)$ *(or* $\mathbb{E}\, \sigma_A(G_k)$*) converges to zero linearly.*

**Remark 2.5** *Note that the complexity in Eq.* (11) *is* $O(n^3)$ *because we have multiply-add operations with (unknown)* $A^{-1}$. *Hence we **do not** apply this greedy strategy in practice, but view it as a theoretical result similar to the random strategy. Moreover, the random method is still practical, and we show the efficiency of our scaled direction compared to the original direction in experiments.*

Finally, we can employ an efficient way (with complexity $O(n^2)$) for updating $L_k$. The main idea is employing the rank-one update and the one-row appending update of QR decomposition with the inverse BFGS update. In Proposition 2.6, we show how to compute $L_{k+1}$ from $L_k$ with $O(n^2)$ flops.

**Proposition 2.6** *Suppose we already have* $H_k \triangleq G_k^{-1} = L_k^\top L_k$, *where* $L_k$ *is an upper triangular matrix. Now we construct* $L_{k+1}$ *with* $O(n^2)$ *flops.*

*Step 1: Using the formulation in Golub and Van Loan [12, Section 12.5.1], we can obtain* $QR$ *decomposition of*

$$L_k \left(I_k - \frac{A u_k u_k^\top}{u_k^\top A u_k}\right) = L_k - \frac{L_k (A u_k)}{u_k^\top A u_k} u_k^\top$$

*with* $O(n^2)$ *flops because it is a rank-one change of* $L_k$.

*Step 2: We have* $L_k \left(I_n - \frac{A u_k u_k^\top}{u_k^\top A u_k}\right) = Q_k R_k$ *from Step 1, with an orthogonal matrix* $Q_k \in \mathbb{R}^{n \times n}$ *and an upper triangular matrix* $R_k \in \mathbb{R}^{n \times n}$. *Denoting* $v_k = \frac{u_k}{\sqrt{u_k^\top A u_k}}$, *we can write*

$$H_{k+1} \overset{(16)}{=} R_k^\top Q_k^\top Q_k R_k + \frac{u_k u_k^\top}{u_k^\top A u_k} = R_k^\top R_k + v_k v_k^\top = \begin{pmatrix} v_k & R_k^\top \end{pmatrix} \begin{pmatrix} v_k^\top \\

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

# A Missing Proofs

## A.1 Proof of Theorem 2.3

Proof: Let $G_{k+1} = \text{SR1}(G_k, A, u_k)$, then

$$(G_{k+1} - A)u_k \overset{(2)}{=} (G_k - A)u_k - \frac{(G_k - A)u_k u_k^\top (G_k - A)}{u_k^\top (G_k - A)u_k} u_k = 0.$$

Moreover, for any $v$ such that $(G_k - A)v = 0$, we have

$$(G_{k+1} - A)v = (G_k - A)v - \frac{(G_k - A)u_k u_k^\top (G_k - A)}{u_k^\top (G_k - A)u_k} v = 0.$$

Therefore, we obtain

$$(G_k - A)\, u_j = 0, \forall j < k. \tag{24}$$

For the *greedy method*, we denote $\bar{u}_k = \bar{u}_A(G_k)$, then from Eq. (24), we can assume $\bar{u}_k \neq \bar{u}_j, \forall k > j$, and $\text{rank}(G_k - A) \leq n - k$. At step $k$, without loss of generality, we assume $\bar{u}_i = e_i, i = 0, \ldots, k-1$. Then $(G_k - A)e_i = 0, \forall i \leq k-1$, leading to $(G_k - A)_{ii} = 0, \forall i \leq k-1$. From Cauchy–Schwarz inequality and $G_k \succeq A$, we obtain

$$\frac{\bar{u}_k^\top (G_k - A)^2 \bar{u}_k}{\bar{u}_k^\top (G_k - A)\bar{u}_k} \geq \frac{\bar{u}_k^\top (G_k - A)\bar{u}_k}{\bar{u}_k^\top \bar{u}_k} \overset{(7)}{=} \max_{1 \leq i \leq n}(G_k - A)_{ii} \geq \frac{1}{n-k}\text{tr}(G_k - A),$$

where the last inequality uses $\text{rank}(G_k - A) \leq n - k$.

Therefore, the greedy choice of $u_k$ leads to

$$\tau_A(G_{k+1}) \leq \left(1 - \frac{1}{n-k}\right)\tau_A(G_k).$$

Consequently, we have

$$\tau_A(G_k) \leq \left(1 - \frac{1}{n-k+1}\right)\tau_A(G_{k-1}) \leq \cdots \leq \left[\prod_{j=1}^{k}\left(1 - \frac{1}{n-j+1}\right)\right]\tau_A(G_0)$$

$$= \left[\prod_{j=1}^{k}\frac{n-j}{n-j+1}\right]\tau_A(G_0) = \frac{n-k}{n}\tau_A(G_0) = \left(1 - \frac{k}{n}\right)\tau_A(G_0).$$

For the *random method* at step $k$, since $G_k \succeq A$, we have $\lambda_i \triangleq \lambda_i(G_k - A) \geq 0, \forall i \in \{1, \ldots, n\}$. We let $r_k \triangleq \text{rank}(G_k - A)$ and denote $G_k - A = U\Lambda U^\top$ as the spectral decomposition of $G_k - A$ with an orthogonal matrix $U$ and a diagonal matrix $\Lambda = \text{diag}\{\lambda_1, \ldots, \lambda_n\}$. Let $v = (v_1, \ldots, v_n)^\top := U^\top u$. We can show in expectation that

$$\mathbb{E}_u \frac{u^\top (G_k - A)^2 u}{u^\top (G_k - A)u} = \mathbb{E}_v \frac{\sum_{i=1}^{r_k}\lambda_i^2 v_i^2}{\sum_{i=1}^{r_k}\lambda_i v_i^2} \geq \mathbb{E}_v \frac{\sum_{i=1}^{r_k}\lambda_i v_i^2}{\sum_{i=1}^{r_k}v_i^2} = \sum_{i=1}^{r_k}\lambda_i \mathbb{E}_v \frac{v_i^2}{\sum_{j=1}^{r_k}v_j^2} = \frac{1}{r_k}\tau_A(G_k).$$

The first equality holds due to $(G_k - A)^2 = U\Lambda^2 U^\top$, and the inequality holds due to the Cauchy–Schwarz inequality and $G_k \succeq A$:

$$\left(\sum_{i=1}^{r_k}\lambda_i^2 v_i^2\right)\left(\sum_{i=1}^{r_k}v_i^2\right) \geq \left(\sum_{i=1}^{r_k}\lambda_i v_i^2\right)^2;$$

the second equality uses the fact that $v$ is still spherically symmetric, thus also permutation invariant:

$$\mathbb{E}_v \frac{v_1^2}{\sum_{j=1}^{r_k}v_j^2} = \cdots = \mathbb{E}_v \frac{v_{r_k}^2}{\sum_{j=1}^{r_k}v_j^2} = \frac{1}{r_k}\sum_{i=1}^{r_k}\mathbb{E}_v \frac{v_i^2}{\sum_{j=1}^{r_k}v_j^2} = \frac{1}{r_k}.$$

Therefore, the random choice of $\boldsymbol{u}_k$ leads to

$$\mathbb{E}_{\boldsymbol{u}_k}\left[\tau_{\boldsymbol{A}}(\boldsymbol{G}_{k+1})|\boldsymbol{G}_k\right] \leq \left(1 - \frac{1}{r_k}\right)\tau_{\boldsymbol{A}}(\boldsymbol{G}_k).$$

Note that $\boldsymbol{u}_0,\ldots,\boldsymbol{u}_{n-1}$ are linearly independent almost surely [23], thus $r_k = \mathrm{rank}(\boldsymbol{G}_k - \boldsymbol{A}) \leq n - k, a.s.$ We finally obtain

$$\mathbb{E}\,\tau_{\boldsymbol{A}}(\boldsymbol{G}_k) \leq \left(1 - \frac{1}{n-k+1}\right)\mathbb{E}\,\tau_{\boldsymbol{A}}(\boldsymbol{G}_{k-1}) \leq \cdots \leq \prod_{j=1}^{k}\left(1 - \frac{1}{n-j+1}\right)\tau_{\boldsymbol{A}}(\boldsymbol{G}_0)$$

$$= \left(1 - \frac{k}{n}\right)\tau_{\boldsymbol{A}}(\boldsymbol{G}_0).$$

$\square$

**Remark A.1** *In fact, for the random method, when applied to $\sigma_{\boldsymbol{A}}(\cdot)$, we could not handle the expectation of the division of random variables with known expectations, because the random variables themselves are non-independent as Rodomanov and Nesterov [19] mentioned. We take $\tau_{\boldsymbol{A}}(\cdot)$ instead, which overcomes this difficulty due to the same spectral decomposition of $(\boldsymbol{G} - \boldsymbol{A})^2$ and $\boldsymbol{G} - \boldsymbol{A}$.*

### A.2 Proof of Theorem 2.4

Proof: For the *greedy method* at step $k$, since $\boldsymbol{G}_k^{-1} = \boldsymbol{L}_k^\top \boldsymbol{L}_k$, we obtain

$$\max_{\tilde{\boldsymbol{u}}\in\{\boldsymbol{e}_1,\ldots,\boldsymbol{e}_n\}}\frac{\tilde{\boldsymbol{u}}^\top \boldsymbol{L}_k^{-\top}\boldsymbol{A}^{-1}\boldsymbol{L}_k^{-1}\tilde{\boldsymbol{u}}}{\tilde{\boldsymbol{u}}^\top\tilde{\boldsymbol{u}}} = \max_{i\in[n]}\left(\boldsymbol{L}_k^{-\top}\boldsymbol{A}^{-1}\boldsymbol{L}_k^{-1}\right)_{ii} \geq \frac{1}{n}\mathrm{tr}\left(\boldsymbol{L}_k^{-\top}\boldsymbol{A}^{-1}\boldsymbol{L}_k^{-1}\right)$$

$$= \frac{1}{n}\mathrm{tr}\left(\left(\boldsymbol{L}_k^\top \boldsymbol{L}_k\right)^{-1}\boldsymbol{A}^{-1}\right) = \frac{1}{n}\mathrm{tr}(\boldsymbol{G}_k\boldsymbol{A}^{-1}). \tag{25}$$

Therefore, the greedy choice of $\boldsymbol{u}_k = \boldsymbol{L}_k^\top\tilde{\boldsymbol{u}}_k$ leads to

$$\sigma_{\boldsymbol{A}}(\boldsymbol{G}_{k+1}) \stackrel{(10)}{\leq} \sigma_{\boldsymbol{A}}(\boldsymbol{G}_k) - \frac{\boldsymbol{u}_k^\top \boldsymbol{G}_k\boldsymbol{A}^{-1}\boldsymbol{G}_k\boldsymbol{u}_k}{\boldsymbol{u}_k^\top \boldsymbol{G}_k\boldsymbol{u}_k} + 1 = \sigma_{\boldsymbol{A}}(\boldsymbol{G}_k) - \frac{\tilde{\boldsymbol{u}}_k^\top \boldsymbol{L}_k^{-\top}\boldsymbol{A}^{-1}\boldsymbol{L}_k^{-1}\tilde{\boldsymbol{u}}_k}{\tilde{\boldsymbol{u}}_k^\top\tilde{\boldsymbol{u}}_k} + 1$$

$$\stackrel{(25)}{\leq} \sigma_{\boldsymbol{A}}(\boldsymbol{G}_k) - \frac{1}{n}\mathrm{tr}(\boldsymbol{G}_k\boldsymbol{A}^{-1}) + 1 = \left(1 - \frac{1}{n}\right)\sigma_{\boldsymbol{A}}(\boldsymbol{G}_k) \leq \cdots \leq \left(1 - \frac{1}{n}\right)^k\sigma_{\boldsymbol{A}}(\boldsymbol{G}_0).$$

For the *random method*, i.e., $\tilde{\boldsymbol{u}} \sim \mathcal{N}(0, \boldsymbol{I}_n)$ or $\tilde{\boldsymbol{u}} \sim \mathrm{Unif}(\mathcal{S}^{n-1})$. We have that $\mathbb{E}_{\tilde{\boldsymbol{u}}}\frac{\tilde{\boldsymbol{u}}\tilde{\boldsymbol{u}}^\top}{\tilde{\boldsymbol{u}}^\top\tilde{\boldsymbol{u}}} = \frac{1}{n}\boldsymbol{I}_n$. Hence, we obtain

$$\mathbb{E}_{\tilde{\boldsymbol{u}}}\frac{\tilde{\boldsymbol{u}}^\top \boldsymbol{L}_k^{-\top}\boldsymbol{A}^{-1}\boldsymbol{L}_k^{-1}\tilde{\boldsymbol{u}}}{\tilde{\boldsymbol{u}}^\top\tilde{\boldsymbol{u}}} = \mathrm{tr}\left[\boldsymbol{L}_k^{-\top}\boldsymbol{A}^{-1}\boldsymbol{L}_k^{-1}\cdot\mathbb{E}_{\tilde{\boldsymbol{u}}}\frac{\tilde{\boldsymbol{u}}\tilde{\boldsymbol{u}}^\top}{\tilde{\boldsymbol{u}}^\top\tilde{\boldsymbol{u}}}\right]$$

$$= \frac{1}{n}\mathrm{tr}(\boldsymbol{L}_k^{-\top}\boldsymbol{A}^{-1}\boldsymbol{L}_k^{-1}) = \frac{1}{n}\mathrm{tr}(\boldsymbol{G}_k\boldsymbol{A}^{-1}), \tag{26}$$

Therefore, the random choice of $\boldsymbol{u}_k = \boldsymbol{L}_k^\top\tilde{\boldsymbol{u}}_k$ leads to

$$\mathbb{E}_{\boldsymbol{u}_k}\left[\sigma_{\boldsymbol{A}}(\boldsymbol{G}_{k+1})|\boldsymbol{G}_k\right] \stackrel{(10)}{=} \mathbb{E}_{\boldsymbol{u}_k}\sigma_{\boldsymbol{A}}(\boldsymbol{G}_k) - \frac{\boldsymbol{u}_k^\top \boldsymbol{G}_k\boldsymbol{A}^{-1}\boldsymbol{G}_k\boldsymbol{u}_k}{\boldsymbol{u}_k^\top \boldsymbol{G}_k\boldsymbol{u}_k} + 1$$

$$= \mathbb{E}_{\tilde{\boldsymbol{u}}_k}\sigma_{\boldsymbol{A}}(\boldsymbol{G}_k) - \frac{\tilde{\boldsymbol{u}}_k^\top \boldsymbol{L}_k^{-\top}\boldsymbol{A}^{-1}\boldsymbol{L}_k^{-1}\tilde{\boldsymbol{u}}_k}{\tilde{\boldsymbol{u}}_k^\top\tilde{\boldsymbol{u}}_k} + 1$$

$$\stackrel{(26)}{=} \sigma_{\boldsymbol{A}}(\boldsymbol{G}_k) - \frac{1}{n}\mathrm{tr}(\boldsymbol{G}_k\boldsymbol{A}^{-1}) + 1 = \left(1 - \frac{1}{n}\right)\sigma_{\boldsymbol{A}}(\boldsymbol{G}_k),$$

showing that

$$\mathbb{E}\,\sigma_{\boldsymbol{A}}(\boldsymbol{G}_k) = \left(1 - \frac{1}{n}\right)\mathbb{E}\,\sigma_{\boldsymbol{A}}(\boldsymbol{G}_{k-1}) = \cdots = \left(1 - \frac{1}{n}\right)^k\sigma_{\boldsymbol{A}}(\boldsymbol{G}_0).$$

$\square$

## A.3 Proof of Theorem 3.2

Proof: We denote $\eta_k = \left\| A^{-1/2} G_k A^{-1/2} \right\|_2$, then $A \preceq G_k \preceq \eta_k A$. Hence we get

$$
\begin{aligned}
\eta_k - 1 &\leq \left\| A^{-1/2} (G_k - A) A^{-1/2} \right\|_2 \\
&\leq \operatorname{tr}\left( A^{-1/2} (G_k - A) A^{-1/2} \right) \stackrel{(6)}{=} \sigma_A(G_k) \qquad (27) \\
&\leq \frac{\operatorname{tr}(G_k - A)}{\mu} \stackrel{(5)}{=} \frac{\tau_A(G_k)}{\mu}. \qquad (28)
\end{aligned}
$$

1). From Eq. (9), we know that under SR1 update,

$$
\mathbb{E}(\eta_k - 1) \stackrel{(28)}{\leq} \frac{\mathbb{E}\tau_A(G_k)}{\mu} \stackrel{(9)}{\leq} \left( 1 - \frac{k}{n} \right) \frac{\tau_A(G_0)}{\mu}.
$$

By Lemma 3.1, this implies

$$
\mathbb{E}\frac{\lambda_f(x_{k+1})}{\lambda_f(x_k)} \leq \mathbb{E}(\eta_k - 1) \leq \left( 1 - \frac{k}{n} \right) \frac{\operatorname{tr}(G_0 - A)}{\mu}.
$$

2). From Eq. (13), we know that under BFGS update,

$$
\mathbb{E}(\eta_k - 1) \stackrel{(27)}{\leq} \mathbb{E}\,\sigma_A(G_k) \stackrel{(13))}{\leq} \left( 1 - \frac{1}{n} \right)^k \sigma_A(G_0).
$$

By Lemma 3.1, this implies

$$
\mathbb{E}\frac{\lambda_f(x_{k+1})}{\lambda_f(x_k)} \leq \mathbb{E}(\eta_k - 1) \leq \left( 1 - \frac{1}{n} \right)^k \operatorname{tr}\left[ (G_0 - A) A^{-1} \right].
$$

$\square$

## A.4 Proof of Theorem 4.2

We need several Lemmas from Rodomanov and Nesterov [19].

**Lemma A.2 (Rodomanov and Nesterov [19] Lemma 4.2)** *Let $r \triangleq \|y - x\|_x$ for $x, y \in \mathbb{R}^n$. Then*

$$
\frac{\nabla^2 f(x)}{1 + Mr} \preceq \nabla^2 f(y) \preceq (1 + Mr) \nabla^2 f(x). \qquad (29)
$$

*Also, for $J \triangleq \int_0^1 \nabla^2 f(x + t(y - x))dt$ and $v \in \{x, y\}$, we have*

$$
\frac{1}{1 + \frac{Mr}{2}} \nabla^2 f(v) \preceq J \preceq \left( 1 + \frac{Mr}{2} \right) \nabla^2 f(v).
$$

**Lemma A.3 (Rodomanov and Nesterov [19] Lemma 4.3 and Lemma 4.4)** *Let $x \in \mathbb{R}^n$, and a symmetric matrix $G$, such that $\nabla^2 f(x) \preceq G \preceq \eta \nabla^2 f(x)$, for some $\eta \geq 1$. Let $x_+ \in \mathbb{R}^n$, and $r = \|x_+ - x\|_x$. Then*

$$
\tilde{G} \triangleq (1 + Mr) G \succeq \nabla^2 f(x_+),
$$

*and for all $u \in \mathbb{R}^n$ and $\tau \in [0, 1]$, we have*

$$
\nabla^2 f(x_+) \preceq \operatorname{Broyd}_\tau\left( \tilde{G}, \nabla^2 f(x_+), u \right) \preceq \left[ (1 + Mr)^2 \eta \right] \nabla^2 f(x_+).
$$

*More specifically, if $x_+ = x - G^{-1}\nabla f(x)$, and letting $\lambda \triangleq \lambda_f(x)$ be such that $M\lambda \leq 2$, then, $r \leq \lambda$, and*

$$
\lambda_f(x_+) \leq \left( 1 + \frac{M\lambda}{2} \right) \frac{\eta - 1 + \frac{M\lambda}{2}}{\eta} \lambda.
$$

**Lemma A.4 (Modified from Rodomanov and Nesterov [19] Lemma 4.8)** *Let* $\boldsymbol{x} \in \mathbb{R}^n$, *and a symmetric matrix* $\boldsymbol{G}$, *such that* $\nabla^2 f(\boldsymbol{x}) \preceq \boldsymbol{G}$. *Let* $\boldsymbol{x}_+ \in \mathbb{R}^n$, *and* $r = \|\boldsymbol{x}_+ - \boldsymbol{x}\|_{\boldsymbol{x}}$, $\tilde{\boldsymbol{G}} = (1 + Mr)\,\boldsymbol{G}$, $\tilde{\boldsymbol{G}}^{-1} = \tilde{\boldsymbol{L}}^\top \tilde{\boldsymbol{L}}$, *and* $\boldsymbol{u}$ *follows from Algorithm 2 applied with* $\tilde{\boldsymbol{G}}, \nabla^2 f(\boldsymbol{x}_+)$. *Then we have*

$$\mathbb{E}_{\tilde{\boldsymbol{u}}} \sigma_{\boldsymbol{x}_+}(\mathrm{BFGS}(\tilde{\boldsymbol{G}}, \nabla^2 f(\boldsymbol{x}_+), \tilde{\boldsymbol{L}}^\top \tilde{\boldsymbol{u}})) \leq \left(1 - \frac{1}{n}\right)(1 + Mr)^2 \left(\sigma_{\boldsymbol{x}}(\boldsymbol{G}) + \frac{2nMr}{1 + Mr}\right),$$

Proof: We already know from Lemma A.3 that $\nabla^2 f(\boldsymbol{x}_+) \preceq \tilde{\boldsymbol{G}}$. Then by Theorem 2.4, we have

$$\mathbb{E}_{\tilde{\boldsymbol{u}}} \sigma_{\boldsymbol{x}_+}(\mathrm{BFGS}(\tilde{\boldsymbol{G}}, \nabla^2 f(\boldsymbol{x}_+), \tilde{\boldsymbol{L}}^\top \tilde{\boldsymbol{u}})) \leq \left(1 - \frac{1}{n}\right) \sigma_{\boldsymbol{x}_+}(\tilde{\boldsymbol{G}}),$$

The remaining proof is same as [19, Lemma 4.8]. To make the paper self-contained, we also show the detail below.

$$
\begin{aligned}
\sigma_{\boldsymbol{x}_+}(\tilde{\boldsymbol{G}}) &\overset{(6)}{=} \left\langle \nabla^2 f(\boldsymbol{x}_+)^{-1}, \tilde{\boldsymbol{G}} \right\rangle - n = (1 + Mr) \left\langle \nabla^2 f(\boldsymbol{x}_+)^{-1}, \boldsymbol{G} \right\rangle - n \\
&\overset{(29)}{\leq} (1 + Mr)^2 \left\langle \nabla^2 f(\boldsymbol{x})^{-1}, \boldsymbol{G} \right\rangle - n \overset{(6)}{=} (1 + Mr)^2 \left(\sigma_{\boldsymbol{x}}(\boldsymbol{G}) + n\right) - n \\
&= (1 + Mr)^2 \sigma_{\boldsymbol{x}}(\boldsymbol{G}) + n\left((1 + Mr)^2 - 1\right) \\
&= (1 + Mr)^2 \sigma_{\boldsymbol{x}}(\boldsymbol{G}) + 2nMr\left(1 + \frac{Mr}{2}\right) \\
&\leq (1 + Mr)^2 \left(\sigma_{\boldsymbol{x}}(\boldsymbol{G}) + \frac{2nMr}{1 + Mr}\right).
\end{aligned}
$$

$\square$

Now we turn to the main proof of Theorem 4.2.

Proof: The derivation is same as [19, Theorem 4.9] by using Lemma A.4. Denote $\lambda_k \triangleq \lambda_f(\boldsymbol{x}_k)$, $\sigma_k \triangleq \sigma_{\boldsymbol{x}_k}(\boldsymbol{G}_k)$ for $k \geq 0$. In view of [19, Theorem 4.7], shown in Appendix B.1, since the initial condition $\frac{\ln 2}{4\kappa(2n+1)} \leq \frac{\ln \frac{3}{2}}{4\kappa}$, the first relation in Eq. (19) is indeed true, and also

$$M \sum_{i=0}^{k} \lambda_i \overset{(51)}{\leq} M\lambda_0 \sum_{i=0}^{k} \left(1 - \frac{\mu}{2L}\right)^i \leq \frac{2L}{\mu} M\lambda_0 \overset{(18)}{\leq} \frac{\ln 2}{2(2n+1)}. \tag{30}$$

Denote $\theta_k := \sigma_k + 2nM\lambda_k$. Let us show by induction that for all $k \geq 0$, we have

$$\mathbb{E}\theta_k \leq \left(1 - \frac{1}{n}\right)^k \frac{2nL}{\mu}. \tag{31}$$

Indeed, since $\boldsymbol{G}_0 = L\boldsymbol{I}_n$ and $\mu\boldsymbol{I}_n \leq \nabla^2 f(\boldsymbol{x}_0) \leq L\boldsymbol{I}_n$, we have

$$
\begin{aligned}
\theta_0 = \sigma_0 + 2nM\lambda_0 &\overset{(6)}{=} \mathrm{tr}\left(\nabla^2 f(\boldsymbol{x}_0)^{-1} \boldsymbol{G}_0\right) - n + 2nM\lambda_0 \\
&\leq \mathrm{tr}\left(\nabla^2 f(\boldsymbol{x}_0)^{-1} \cdot \frac{L}{\mu}\nabla^2 f(\boldsymbol{x}_0)\right) - n + 2nM\lambda_0 = n\left(\frac{L}{\mu} - 1\right) + \frac{n\ln 2}{2\kappa(2n+1)} \leq \frac{nL}{\mu}.
\end{aligned} \tag{32}
$$

Therefore, for $k = 0$, Eq. (31) is satisfied. Now suppose that it is also satisfied for some $k \geq 0$. Since $\nabla^2 f(\boldsymbol{x}_k) \preceq \boldsymbol{G}_k$, we know that

$$\boldsymbol{G}_k - \nabla^2 f(\boldsymbol{x}_k) \preceq \sigma_k \nabla^2 f(\boldsymbol{x}_k),$$

or equivalently,

$$\boldsymbol{G}_k \preceq (1 + \sigma_k)\,\nabla^2 f(\boldsymbol{x}_k).$$

Since Eq. (30) shows $M\lambda_k \leq 2$, we can apply Lemma A.3, leading to

$$r_k \triangleq \|\boldsymbol{x}_{k+1} - \boldsymbol{x}_k\|_{\boldsymbol{x}_k} \leq \lambda_k, \tag{33}$$

and

$$\lambda_{k+1} \leq \left(1 + \frac{M\lambda_k}{2}\right) \frac{\sigma_k + \frac{M\lambda_k}{2}}{1 + \sigma_k} \lambda_k \leq \left(1 + \frac{M\lambda_k}{2}\right) (\sigma_k + 2nM\lambda_k) \lambda_k \tag{34}$$
$$\leq e^{2M\lambda_k} (\sigma_k + 2nM\lambda_k) \lambda_k.$$

Further, by Lemma A.4, we have

$$\begin{aligned}
\mathbb{E}_{\boldsymbol{u}_k} \sigma_{k+1} &\leq \left(1 - \frac{1}{n}\right)(1 + Mr_k)^2 \left(\sigma_k + \frac{2nMr_k}{1 + Mr_k}\right) \\
&\overset{(33)}{\leq} \left(1 - \frac{1}{n}\right)(1 + M\lambda_k)^2 (\sigma_k + 2nM\lambda_k) \\
&\leq \left(1 - \frac{1}{n}\right) e^{2M\lambda_k} (\sigma_k + 2nM\lambda_k).
\end{aligned}$$

Note that $\frac{1}{2} \leq 1 - \frac{1}{n}$ because $n \geq 2$, and take expectation for all randomness. We obtain

$$\begin{aligned}
&\mathbb{E}\left[\sigma_{k+1} + 2nM\lambda_{k+1}\right] \\
&\leq \mathbb{E}\left(1 - \frac{1}{n}\right) e^{2M\lambda_k} (\sigma_k + 2nM\lambda_k) + e^{2M\lambda_k} 2nM (\sigma_k + 2nM\lambda_k) \lambda_k \\
&\leq \mathbb{E}\left(1 - \frac{1}{n}\right) e^{2M\lambda_k} (\sigma_k + 2nM\lambda_k) + \left(1 - \frac{1}{n}\right) e^{2M\lambda_k} 4nM (\sigma_k + 2nM\lambda_k) \lambda_k \\
&= \mathbb{E}\left(1 - \frac{1}{n}\right) e^{2M\lambda_k} (1 + 4nM\lambda_k)(\sigma_k + 2nM\lambda_k) \\
&\leq \mathbb{E}\left(1 - \frac{1}{n}\right) e^{2(2n+1)M\lambda_k} (\sigma_k + 2nM\lambda_k) \\
&\overset{(51)}{\leq} \mathbb{E}\left(1 - \frac{1}{n}\right) e^{2(2n+1)M\lambda_0\left(1 - \frac{\mu}{2L}\right)^k} (\sigma_k + 2nM\lambda_k).
\end{aligned}$$

Therefore

$$\begin{aligned}
\mathbb{E}\theta_{k+1} &\leq \left(1 - \frac{1}{n}\right) e^{2(2n+1)M\lambda_0\left(1 - \frac{\mu}{2L}\right)^k} \mathbb{E}\theta_k \\
&\leq \left(1 - \frac{1}{n}\right)^{k+1} e^{2(2n+1)M\lambda_0 \sum_{i=0}^{k}\left(1 - \frac{\mu}{2L}\right)^i} \mathbb{E}\theta_0 \tag{35} \\
&\leq \left(1 - \frac{1}{n}\right)^{k+1} e^{4\kappa(2n+1)M\lambda_0} \theta_0 \\
&\overset{\substack{(18)\\(32)}}{\leq} \left(1 - \frac{1}{n}\right)^{k+1} 2n\kappa. \tag{36}
\end{aligned}$$

Thus, Eq. (31) is proved.

Let us fix some $k \geq 0$. Since $\lambda_k \geq 0$, we have

$$\mathbb{E}\sigma_k \leq \mathbb{E}\sigma_k + 2nM\lambda_k = \mathbb{E}\theta_k \overset{(31)}{\leq} \left(1 - \frac{1}{n}\right)^k 2n\kappa.$$

This proves the second relation in Eq. (19). Finally,

$$\begin{aligned}
\mathbb{E}\lambda_{k+1}/\lambda_k &\overset{(34)}{\leq} \mathbb{E}e^{2M\lambda_k}\theta_k \leq \mathbb{E}e^{2(2n+1)M\lambda_k}\theta_k \overset{(51)}{\leq} e^{2(2n+1)M\lambda_0\left(1 - \frac{\mu}{2L}\right)^k} \mathbb{E}\theta_k \\
&\overset{(35)}{\leq} \left(1 - \frac{1}{n}\right)^k e^{2(2n+1)M\lambda_0 \sum_{i=0}^{k}\left(1 - \frac{\mu}{2L}\right)^i} \theta_0 \overset{\substack{(18),(32)}}{\leq} \left(1 - \frac{1}{n}\right)^k 2n\kappa.
\end{aligned}$$

Thus, Eq. (20) is proved. $\qquad\square$

## A.5 Proof of Theorem 4.3

Proof: We denote the sequence $\{\eta_k\}$ that satisfies

$$\text{tr}\left(\boldsymbol{G}_k - \nabla^2 f(\boldsymbol{x}_k)\right) = \eta_k \,\text{tr}\left(\nabla^2 f(\boldsymbol{x}_k)\right), \tag{37}$$

and

$$\lambda_k \triangleq \lambda_f(\boldsymbol{x}_{k+1}), r_k \triangleq \|\boldsymbol{x}_{k+1} - \boldsymbol{x}_k\|_{\boldsymbol{x}_k}, \sigma_k \triangleq \sigma_{\nabla^2 f(\boldsymbol{x}_k)}(\boldsymbol{G}_k).$$

From Theorem 2.3, we obtain

$$\begin{aligned}
\mathbb{E}_{\boldsymbol{u}_k} \,\text{tr}\left(\boldsymbol{G}_{k+1} - \nabla^2 f(\boldsymbol{x}_{k+1})\right) &\leq \left(1 - \frac{1}{n}\right) \text{tr}\left(\tilde{\boldsymbol{G}}_k - \nabla^2 f(\boldsymbol{x}_{k+1})\right) \\
&\leq \left(1 - \frac{1}{n}\right) \text{tr}\left((1 + Mr_k)\,\boldsymbol{G}_k - \frac{1}{1 + Mr_k}\nabla^2 f(\boldsymbol{x}_k)\right) \\
&\stackrel{(37)}{=} \left(1 - \frac{1}{n}\right) \left((1 + Mr_k)(1 + \eta_k) - \frac{1}{1 + Mr_k}\right) \text{tr}\left(\nabla^2 f(\boldsymbol{x}_k)\right) \\
&\leq \left(1 - \frac{1}{n}\right) \left((1 + Mr_k)^2 (1 + \eta_k) - 1\right) \text{tr}\left(\nabla^2 f(\boldsymbol{x}_{k+1})\right),
\end{aligned}$$

where the first inequality uses Theorem 2.3 and the second inequality uses Lemma A.2. The last inequality uses Lemma A.2 again.

We can apply [19, Theorem 4.7], shown in Appendix B.1, since the initial condition $\frac{\ln 2}{4\kappa(1+2n\kappa)} \leq \frac{\ln\frac{3}{2}}{4\kappa}$, and our algorithm can be viewed as a specific version of theirs. Hence $M\lambda_k \leq \left(1 - \frac{\mu}{2L}\right)^k M\lambda_0 < 2$, and using Lemma A.3, we obtain $r_k \leq \lambda_k$. Therefore

$$\begin{aligned}
\mathbb{E}_{\boldsymbol{u}_k} \eta_{k+1} &\leq \left(1 - \frac{1}{n}\right) \left((1 + Mr_k)^2 (1 + \eta_k) - 1\right) \\
&\leq \left(1 - \frac{1}{n}\right) (1 + Mr_k)^2 \left(\eta_k + 1 - \frac{1}{(1 + Mr_k)^2}\right) \\
&\leq \left(1 - \frac{1}{n}\right) (1 + Mr_k)^2 \left(\eta_k + \frac{2Mr_k + (Mr_k)^2}{(1 + Mr_k)^2}\right) \\
&\leq \left(1 - \frac{1}{n}\right) (1 + Mr_k)^2 \left(\eta_k + 2Mr_k\right) \\
&\leq \left(1 - \frac{1}{n}\right) (1 + M\lambda_k)^2 \left(\eta_k + 2M\lambda_k\right).
\end{aligned} \tag{38}$$

Moreover, since $\mu\boldsymbol{I}_n \preceq \nabla^2 f(\boldsymbol{x}) \preceq L\boldsymbol{I}_n$, we have

$$\sigma_k = \text{tr}\left((\boldsymbol{G}_k - \nabla^2 f(\boldsymbol{x}_k))\,\nabla^2 f(\boldsymbol{x}_k)^{-1}\right) \leq \frac{1}{\mu}\text{tr}\left(\boldsymbol{G}_k - \nabla^2 f(\boldsymbol{x}_k)\right) \stackrel{(37)}{=} \frac{\eta_k}{\mu}\text{tr}\left(\nabla^2 f(\boldsymbol{x}_k)\right) \leq \eta_k n\kappa. \tag{39}$$

Applying Lemma A.3 and noting that $\boldsymbol{G}_k \preceq (1 + \sigma_k)\,\nabla^2 f(\boldsymbol{x}_k)$, we get

$$\lambda_{k+1} \leq \left(1 + \frac{M\lambda_k}{2}\right) \frac{\sigma_k + \frac{M\lambda_k}{2}}{1 + \sigma_k}\lambda_k \stackrel{(39)}{\leq} \left(1 + \frac{M\lambda_k}{2}\right) \left(\eta_k n\kappa + \frac{M\lambda_k}{2}\right)\lambda_k. \tag{40}$$

Combining Eq. (40) and Eq. (38), we obtain

$\mathbb{E}\,\eta_{k+1} + 2M\lambda_{k+1}$

$$\leq \mathbb{E}\left(1 - \frac{1}{n}\right)(1 + M\lambda_k)^2(\eta_k + 2M\lambda_k) + \left(1 + \frac{M\lambda_k}{2}\right)\left(\eta_k n\kappa + \frac{M\lambda_k}{2}\right)2M\lambda_k$$

$$\leq \mathbb{E}\left(1 - \frac{1}{n}\right)(1 + M\lambda_k)^2(\eta_k + 2M\lambda_k) + (1 + M\lambda_k)(\eta_k + 2M\lambda_k)2n\kappa M\lambda_k$$

$$\leq \mathbb{E}\left(1 - \frac{1}{n}\right)(1 + M\lambda_k)^2(\eta_k + 2M\lambda_k) + \left(1 - \frac{1}{n}\right)(1 + M\lambda_k)^2(\eta_k + 2M\lambda_k)4n\kappa M\lambda_k$$

$$\leq \mathbb{E}\left(1 - \frac{1}{n}\right)(1 + M\lambda_k)^2(\eta_k + 2M\lambda_k)(4n\kappa M\lambda_k + 1)$$

$$\leq \mathbb{E}\left(1 - \frac{1}{n}\right)e^{2M\lambda_k + 4n\kappa M\lambda_k}(\eta_k + 2M\lambda_k),$$

(41)

where the third inequality uses $1 \leq 2\left(1 - \frac{1}{n}\right)$ because $n \geq 2$. Now we define

$$\theta_{k+1} \triangleq \eta_k + 2M\lambda_k.$$

In the following, we prove by induction that

$$\mathbb{E}\theta_k \leq \left(1 - \frac{1}{n}\right)^k 2\kappa. \tag{42}$$

1) From the initial condition: $M\lambda_0 \leq \frac{\ln 2}{4\kappa(1 + 2n\kappa)}$, and note that $\boldsymbol{G}_0 \preceq \kappa\nabla^2 f(\boldsymbol{x}_0)$, thus

$$\theta_0 = \eta_0 + 2M\lambda_0 \leq \kappa - 1 + \frac{\ln 2}{2\kappa(1 + 2n\kappa)} \leq \kappa. \tag{43}$$

2) For $k \geq 1$, from Eq. (41), we obtain

$$
\begin{aligned}
\mathbb{E}\theta_{k+1} \quad &\leq \quad \mathbb{E}\left(1 - \frac{1}{n}\right)e^{(2+4n\kappa)M\lambda_k}\theta_k \\
&\overset{(51)}{\leq} \quad \left(1 - \frac{1}{n}\right)e^{(2+4n\kappa)M\lambda_0\left(1 - \frac{\mu}{2L}\right)^k}\mathbb{E}\theta_k \\
&\leq \quad \left(1 - \frac{1}{n}\right)^{k+1}e^{(2+4n\kappa)M\lambda_0\sum_{i=0}^{k}\left(1 - \frac{\mu}{2L}\right)^i}\theta_0 \\
&\leq \quad \left(1 - \frac{1}{n}\right)^{k+1}e^{2\kappa(2+4n\kappa)M\lambda_0}\theta_0 \\
&\overset{(21),(43)}{\leq} \quad \left(1 - \frac{1}{n}\right)^{k+1}2\kappa.
\end{aligned}
$$
(44)

(45)

Combining 1) and 2), Eq. (42) holds for all $k \geq 0$. Therefore,

$$\mathbb{E}\eta_k \leq \mathbb{E}\theta_k \leq \left(1 - \frac{1}{n}\right)^k 2\kappa,$$

and

$$\mathbb{E}\sigma_k \overset{(39)}{\leq} \mathbb{E}\,\eta_k n\kappa \leq 2n\kappa^2\left(1 - \frac{1}{n}\right)^k.$$

This proves the second relation in Eq. (22). Finally, from Eq. (40),

$$
\begin{aligned}
\mathbb{E}\lambda_{k+1}/\lambda_k \quad &\overset{(40)}{\leq} \quad \mathbb{E}e^{2M\lambda_k}n\kappa\theta_k \leq \mathbb{E}e^{(2+4n\kappa)M\lambda_k}n\kappa\theta_k \overset{(51)}{\leq} e^{(2+4n\kappa)M\lambda_0\left(1 - \frac{\mu}{2L}\right)^k}n\kappa\mathbb{E}\theta_k \\
&\overset{(44)}{\leq} \quad \left(1 - \frac{1}{n}\right)^k n\kappa e^{(2+4n\kappa)M\lambda_0\sum_{i=0}^{k}\left(1 - \frac{\mu}{2L}\right)^i}\theta_0 \\
&\overset{(21),(43)}{\leq} \quad \left(1 - \frac{1}{n}\right)^k 2n\kappa^2.
\end{aligned}
$$

Thus, Eq. (23) is proved. □

### A.6 Proof of Corollary 4.4

We first give a probabilistic perspective of Theorem 4.2, particularly for the random method.

**Corollary A.5** *Suppose that in Algorithm 4, the random BFGS update is used, and that the initial point $\boldsymbol{x}_0$ is sufficiently close to the solution:*

$$M\lambda_f(\boldsymbol{x}_0) \leq \frac{\ln 2}{4\kappa(2n+1)}. \tag{46}$$

*Then with probability $1 - \delta$, we have*

$$\lambda_f(\boldsymbol{x}_{k+1}) \leq \frac{2n^3\kappa}{\delta}\left(1 - \frac{1}{n+1}\right)^k \lambda_f(\boldsymbol{x}_k), \text{ for all } k \in \mathbb{N}. \tag{47}$$

Proof: Note that $\lambda_f(\boldsymbol{x}_k) \geq 0$. Using Markov's inequality, we have

$$\mathbb{P}\left(\frac{\lambda_f(\boldsymbol{x}_{k+1})}{\lambda_f(\boldsymbol{x}_k)} \geq \frac{2n\kappa}{\epsilon_k}\left(1 - \frac{1}{n}\right)^k\right) \leq \frac{\mathbb{E}\lambda_f(\boldsymbol{x}_{k+1})/\lambda_f(\boldsymbol{x}_k)}{\frac{2n\kappa}{\epsilon_k}\left(1 - \frac{1}{n}\right)^k} \overset{(20)}{\leq} \epsilon_k. \tag{48}$$

Choosing $\epsilon_k = \delta(1 - q)q^k$ for some positive $q < 1$, we have

$$\mathbb{P}\left(\frac{\lambda_f(\boldsymbol{x}_{k+1})}{\lambda_f(\boldsymbol{x}_k)} \geq \frac{2n\kappa}{\epsilon_k}\left(1 - \frac{1}{n}\right)^k, \exists\, k \in \mathbb{N}\right) \leq \sum_{k=0}^{\infty} \mathbb{P}\left(\frac{\lambda_f(\boldsymbol{x}_{k+1})}{\lambda_f(\boldsymbol{x}_k)} \geq \frac{2n\kappa}{\epsilon_k}\left(1 - \frac{1}{n}\right)^k\right)$$

$$\overset{(48)}{\leq} \sum_{k=0}^{\infty}\epsilon_k = \sum_{k=0}^{\infty}\delta(1 - q)q^k = \delta.$$

Therefore, we obtain with probability $1 - \delta$,

$$\lambda_f(\boldsymbol{x}_{k+1}) \leq \left(\frac{1 - \frac{1}{n}}{q}\right)^k \cdot \frac{2n\kappa}{(1 - q)\delta} \cdot \lambda_f(\boldsymbol{x}_k), \forall k \in \mathbb{N}.$$

If we set $q = 1 - 1/n^2$, we could obtain with probability $1 - \delta$,

$$\lambda_f(\boldsymbol{x}_{k+1}) \leq \frac{2n^3\kappa}{\delta}\left(1 + \frac{1}{n}\right)^{-k}\lambda_f(\boldsymbol{x}_k) = \frac{2n^3\kappa}{\delta}\left(1 - \frac{1}{n+1}\right)^k\lambda_f(\boldsymbol{x}_k), \text{ for all } k \in \mathbb{N}.$$

$\square$

Moreover, Theorem 4.3 could also convert to such formulation with the same argument.

**Corollary A.6** *For Algorithm 4 with the random SR1 update, suppose that the initial point $\boldsymbol{x}_0$ is sufficiently close to the solution:*

$$M\lambda_f(\boldsymbol{x}_0) \leq \frac{\ln 2}{4\kappa(2n\kappa + 1)}. \tag{49}$$

*Then with probability $1 - \delta$, we have*

$$\lambda_f(\boldsymbol{x}_{k+1}) \leq \frac{2n^3\kappa^2}{\delta}\left(1 - \frac{1}{n+1}\right)^k\lambda_f(\boldsymbol{x}_k), \forall k \in \mathbb{N}. \tag{50}$$

The proof is totally same as the random BFGS method, so we omit it. Now we turn to the proof of Corollary 4.4.

Proof: 1) For the greedy methods, there is no randomness. We can apply Theorems 4.2 and 4.3 directly.

For the greedy BFGS method, Theorem 4.2 already shows the superlinear convergence rate of $\lambda_f(\boldsymbol{x}_k)$. Now we combine this result with [19, Theorem 4.7] to give the whole period convergence estimator. Denote by $k_1 \geq 0$ the number of the first iteration, for which

$$\left(1 - \frac{1}{2\kappa}\right)^{k_1} \leq \frac{1}{2n+1}.$$

Clearly, $k_1 \leq 2\kappa \ln(2n+1)$.

Since the initial point $\boldsymbol{x}_0$ is sufficiently close to the solution: $M\lambda_f(\boldsymbol{x}_0) \leq \frac{\ln \frac{3}{2}}{4\kappa}$, and reusing the result of [19, Theorem 4.7], we obtain

$$M\lambda_f(\boldsymbol{x}_{k_1}) \leq M\left(1 - \frac{1}{2\kappa}\right)^{k_1} \lambda_f(\boldsymbol{x}_0) \leq \frac{\ln 2}{4\kappa(2n+1)},$$

which satisfies the initial condition in Theorem 4.2.

Denote by $k_2 \geq 0$ the number of the first iteration, for which

$$\left(1 - \frac{1}{n}\right)^{k_2} 2n\kappa \leq \frac{1}{2}.$$

Clearly, $k_2 \leq n\ln(4n\kappa)$.

By Theorem 4.2, for all $k \geq 0$, we have

$$\lambda_f(\boldsymbol{x}_{k_1+k_2+k+1}) \leq \left(1 - \frac{1}{n}\right)^{k_2+k} 2n\kappa\,\lambda_f(\boldsymbol{x}_{k_1+k_2+k}) \leq \left(1 - \frac{1}{n}\right)^{k} \frac{1}{2}\lambda_f(\boldsymbol{x}_{k_1+k_2+k}).$$

Therefore,

$$\lambda_f(\boldsymbol{x}_{k_1+k_2+k}) \leq \left(1 - \frac{1}{n}\right)^{k(k-1)/2} \left(\frac{1}{2}\right)^{k} \lambda_f(\boldsymbol{x}_{k_1+k_2}),$$

and

$$\lambda_f(\boldsymbol{x}_{k_1+k_2}) \leq \left(1 - \frac{1}{2\kappa}\right)^{k_1+k_2} \lambda_f(\boldsymbol{x}_0).$$

Finally, choosing $k_0 = k_1 + k_2 = O\left(\max\{n, \kappa\}\ln(n\kappa)\right)$, we obtain

$$\lambda_f(\boldsymbol{x}_{k_0+k}) \leq \left(1 - \frac{1}{n}\right)^{k(k-1)/2} \left(\frac{1}{2}\right)^{k} \left(1 - \frac{1}{2\kappa}\right)^{k_0} \lambda_f(\boldsymbol{x}_0).$$

We can give similar analysis for the greedy SR1 method with $k_1 \leq 2\kappa\ln(2n\kappa + 1)$ and $k_2 \leq n\ln(4n\kappa^2)$. Thus $k_0 = k_1 + k_2 = O\left(\max\{n, \kappa\}\ln(n\kappa)\right)$ as well.

2) For the random methods, we can apply Corollary A.5 and Corollary A.6 to give the rates.

For the random BFGS method, Corollary A.5 already shows the superlinear convergence rate of $\lambda_f(\boldsymbol{x}_k)$. Now we combine this result with [19, Theorem 4.7] to give the whole period convergence estimator. Denote by $k_1 \geq 0$ the number of the first iteration, for which

$$\left(1 - \frac{1}{2\kappa}\right)^{k_1} \leq \frac{1}{2n+1}.$$

Clearly, $k_1 \leq 2\kappa\ln(2n+1)$.

Since the initial point $\boldsymbol{x}_0$ is sufficiently close to the solution: $M\lambda_f(\boldsymbol{x}_0) \leq \frac{\ln \frac{3}{2}}{4\kappa}$, and reusing the result of [19, Theorem 4.7], we obtain

$$M\lambda_f(\boldsymbol{x}_{k_1}) \leq M\left(1 - \frac{1}{2\kappa}\right)^{k_1} \lambda_f(\boldsymbol{x}_0) \leq \frac{\ln 2}{4\kappa(2n+1)},$$

which satisfies the initial condition in Corollary A.5.

Denote by $k_2 \geq 0$ the number of the first iteration, for which

$$\frac{2n^3\kappa}{\delta}\left(1 - \frac{1}{n+1}\right)^{k_2} \leq \frac{1}{2}.$$

Clearly, $k_2 \leq (n+1)\ln(4n^3\kappa/\delta)$.

By Theorem 4.3, for all $k \geq 0$, we have with probability $1 - \delta$ for any $\delta \in (0, 1)$

$$\lambda_f(\boldsymbol{x}_{k_1+k_2+k+1}) \leq \frac{2n^3\kappa}{\delta} \left(1 - \frac{1}{n+1}\right)^{k_2+k} \lambda_f(\boldsymbol{x}_{k_1+k_2+k}) \leq \left(1 - \frac{1}{n+1}\right)^k \cdot \frac{1}{2} \lambda_f(\boldsymbol{x}_{k_1+k_2+k}).$$

Therefore,

$$\lambda_f(\boldsymbol{x}_{k_1+k_2+k}) \leq \left(1 - \frac{1}{n+1}\right)^{k(k-1)/2} \left(\frac{1}{2}\right)^k \lambda_f(\boldsymbol{x}_{k_1+k_2}),$$

and

$$\lambda_f(\boldsymbol{x}_{k_1+k_2}) \leq \left(1 - \frac{1}{2\kappa}\right)^{k_1+k_2} \lambda_f(\boldsymbol{x}_0).$$

Finally, choosing $k_0 = k_1 + k_2 = O\left(\max\{n, \kappa\} \ln(n\kappa/\delta)\right)$, we obtain

$$\lambda_f(\boldsymbol{x}_{k_0+k}) \leq \left(1 - \frac{1}{n+1}\right)^{k(k-1)/2} \left(\frac{1}{2}\right)^k \left(1 - \frac{1}{2\kappa}\right)^{k_0} \lambda_f(\boldsymbol{x}_0).$$

We can give similar analysis for the random SR1 method with $k_1 \leq 2\kappa \ln(2n\kappa + 1)$ and $k_2 \leq (n+1)\ln(4n^3\kappa^2/\delta)$. Thus $k_0 = k_1 + k_2 = O\left(\max\{n, \kappa\} \ln(n\kappa/\delta)\right)$ as well. $\qquad \square$

## B    Auxiliary Theorem

---

**Algorithm 5** Quasi-Newton Method [19, Scheme (4.17)]

---

1: Initialization: Choose $\boldsymbol{x}_0 \in \mathbb{R}^n$. Set $\boldsymbol{G}_0 = L\boldsymbol{I}_n$.
2: **for** $k \geq 0$ **do**
3:     Update $\boldsymbol{x}_{k+1} = \boldsymbol{x}_k - \boldsymbol{G}_k^{-1}\nabla f(\boldsymbol{x}_k)$.
4:     Compute $r_k = \|\boldsymbol{x}_{k+1} - \boldsymbol{x}_k\|_{\boldsymbol{x}_k}$ and set $\tilde{\boldsymbol{G}}_k = (1 + Mr_k)\boldsymbol{G}_k$.
5:     Choose $\boldsymbol{u}_k \in \mathbb{R}^n$ and $\tau_k \in [0, 1]$.
6:     Compute $\boldsymbol{G}_{k+1} = \mathrm{Broyd}_{\tau_k}\left(\tilde{\boldsymbol{G}}_k, \nabla^2 f(\boldsymbol{x}_{k+1}), \boldsymbol{u}_k\right)$.
7: **end for**

---

**Theorem B.1 (Rodomanov and Nesterov [19] Theorem 4.7)** *Under Algorithm 5, suppose the initial point $\boldsymbol{x}_0$ is sufficiently close to the solution:*

$$M\lambda_f(\boldsymbol{x}_0) \leq \frac{\ln\frac{3}{2}}{4\kappa}.$$

*Then, for all $k \geq 0$, we have*

$$\nabla^2 f(\boldsymbol{x}_k) \preceq \boldsymbol{G}_k \preceq e^{2M\sum_{i=0}^{k-1}\lambda_f(\boldsymbol{x}_i)}\frac{L}{\mu}\nabla^2 f(\boldsymbol{x}_k) \preceq \frac{3L}{2\mu}\nabla^2 f(\boldsymbol{x}_k),$$

*and*

$$\lambda_f(\boldsymbol{x}_k) \leq \left(1 - \frac{\mu}{2L}\right)^k \lambda_f(\boldsymbol{x}_0). \tag{51}$$

**Remark B.2** *Note that the choice of $\boldsymbol{u}_k$ and the update rule in Algorithm 5 are arbitrary, thus our algorithms can be viewed as a special case. Therefore, Theorem B.1 always holds as long as the initial point is sufficiently close to the solution.*

## C    Missing Experiments and Detail

### C.1    Condition-number-free Property of Our Modified BFGS Update

For the matrix approximation task in Section 5, we list the variation of $\sigma_{\boldsymbol{A}}(\boldsymbol{G}_k)$ with various condition numbers $\kappa$ shown in Figure 4. We find scaled directions ($\boldsymbol{L}^{\top}\boldsymbol{u}$) have the same convergence rates in different condition numbers. Thus we conclude our scaled directions give a condition-number-free convergence rate as our theory shows. However, directly selecting random directions (RaBFGSv1) converges much slower than the scaled random directions (RaBFGSv2), and the large condition number could cause slow convergence.

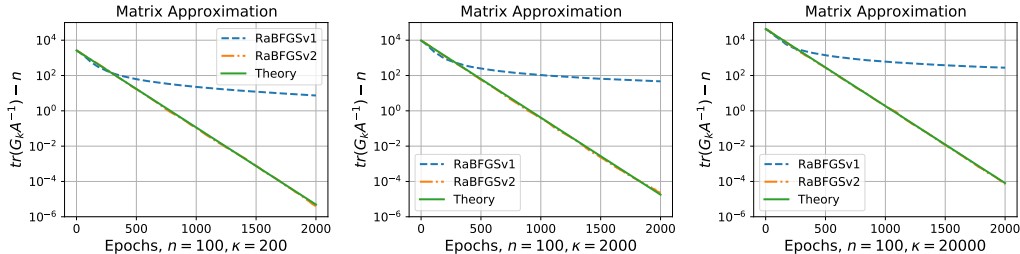

Figure 4: The variation of $\sigma_{\boldsymbol{A}}(\boldsymbol{G}_k)$ during our Random BFGS (RaBFGSv2) update and the original random version (RaBFGSv1) for approximating a matrix $\boldsymbol{A}$ that $\mu\boldsymbol{I}_n \preceq \boldsymbol{A} \preceq L\boldsymbol{I}_n$ with various $\kappa$.

## C.2  Experimental Settings in Section 5

**Regularized Logistic Regression.**    The gradient of function $f$ is

$$\nabla f(\boldsymbol{w}) = -\sum_{i=1}^{N} \frac{e^{y_i \boldsymbol{w}^\top \boldsymbol{x}_i}}{1 + e^{y_i \boldsymbol{w}^\top \boldsymbol{x}_i}} \cdot y_i \boldsymbol{x}_i + \gamma \boldsymbol{w}, \ \boldsymbol{w} \in \mathbb{R}^n,$$

and the Hessian is

$$\nabla^2 f(\boldsymbol{w}) = \sum_{i=1}^{N} \frac{e^{y_i \boldsymbol{w}^\top \boldsymbol{x}_i}}{\left(1 + e^{y_i \boldsymbol{w}^\top \boldsymbol{x}_i}\right)^2} \cdot \boldsymbol{x}_i \boldsymbol{x}_i^\top + \gamma \boldsymbol{I}_n, \ \boldsymbol{w} \in \mathbb{R}^n.$$

Thus $\gamma \boldsymbol{I}_n \preceq \nabla^2 f(\boldsymbol{w}) \preceq L\boldsymbol{I}_n$ with $L = \lambda_{\max}(\boldsymbol{X}\boldsymbol{X}^\top)/4 + \gamma$, and $\kappa = L/\gamma$. In particular, the Hessian-vector product for this function can be computed with the similar complexity of that for computing the gradient, guaranteeing $O(nN)$ complexity in each step as below

$$\nabla^2 f(\boldsymbol{w})\boldsymbol{v} = \sum_{i=1}^{N} \frac{e^{y_i \boldsymbol{w}^\top \boldsymbol{x}_i}}{\left(1 + e^{y_i \boldsymbol{w}^\top \boldsymbol{x}_i}\right)^2} \cdot \left(\boldsymbol{x}_i^\top \boldsymbol{v}\right) \cdot \boldsymbol{x}_i + \gamma \boldsymbol{v}, \ \boldsymbol{w} \in \mathbb{R}^n.$$

We compare $\|\nabla f(\boldsymbol{w}_k)\|$ obtained by different methods.

**Regularized Log-Sum-Exp.**    We need access to the gradient of function $f$:

$$\nabla f(\boldsymbol{x}) = g(\boldsymbol{x}) + \sum_{j=1}^{m} \left(\boldsymbol{c}_j^\top \boldsymbol{x}\right) \boldsymbol{c}_j + \gamma \boldsymbol{x}, g(\boldsymbol{x}) := \sum_{j=1}^{m} \pi_j(\boldsymbol{x})\boldsymbol{c}_j,$$

where

$$\pi_j(\boldsymbol{x}) := \frac{e^{\boldsymbol{c}_j^\top \boldsymbol{x} - b_j}}{\sum_{i=1}^{m} e^{\boldsymbol{c}_i^\top \boldsymbol{x} - b_i}} \in [0, 1], j = 1, \ldots, m.$$

Thus we can see $\nabla f(\boldsymbol{x})$ can be computed in $O(mn)$ operations for a given point $\boldsymbol{x} \in \mathbb{R}^n$. Moreover, we have analytic Hessian and Hessian-vector product expression below:

$$\nabla^2 f(\boldsymbol{x}) = \sum_{j=1}^{m} \left(\pi_j(\boldsymbol{x}) + 1\right) \boldsymbol{c}_j \boldsymbol{c}_j^\top - g(\boldsymbol{x})g(\boldsymbol{x})^\top + \gamma I_n,$$

and for a given direction $\boldsymbol{h} \in \mathbb{R}^n$,

$$\nabla^2 f(\boldsymbol{x})\boldsymbol{h} = \sum_{j=1}^{m} \left(\pi_j(\boldsymbol{x}) + 1\right) \left(\boldsymbol{c}_j^\top \boldsymbol{h}\right) \boldsymbol{c}_j - \left(g(\boldsymbol{x})^\top \boldsymbol{h}\right) g(\boldsymbol{x}) + \gamma \boldsymbol{h}.$$

Hence, the Lipschitz constant of the gradient of $f$ can be taken as $L = 2\lambda_{\max}(CC^\top) + \gamma$, and $\kappa = L/\gamma$. As mentioned in [19, Section 5.1], the strong self-concordancy parameter is $M = 2$. So we apply the *correction strategy* ($\tilde{\boldsymbol{G}}_k = (1 + Mr_k)\boldsymbol{G}_k$).

We also adopt the same synthetic data as [19, Section 5.1]. First, we generate a collection of random vectors $\hat{c}_1, \ldots, \hat{c}_m$ with entries, uniformly distributed in the interval $[-1, 1]$. Then we generate $b_1, \ldots, b_m$ from the same distribution. Using this data, we form a preliminary function

$$\hat{f}(\boldsymbol{x}) := \ln \left( \sum_{j=1}^{m} e^{\hat{c}_j^\top \boldsymbol{x} - b_j} \right),$$

and finally define

$$\boldsymbol{c}_j := \hat{c}_j - \nabla \hat{f}(\boldsymbol{0}), j = 1, \ldots, m.$$

Note that by construction

$$\nabla f(\boldsymbol{0}) = \frac{1}{\sum_{i=1}^{m} e^{-b_i}} \sum_{j=1}^{m} e^{-b_j} \left( \hat{c}_j - \nabla \hat{f}(\boldsymbol{0}) \right) = \boldsymbol{0}.$$

So the unique minimizer of our test function is $\boldsymbol{x}^* = \boldsymbol{0}$. The starting point $\boldsymbol{x}_0$ for all methods is the same and generated randomly from the uniform distribution on the standard Euclidean sphere of radius $1/n$ centered at the minimizer, i.e., $\boldsymbol{x}_0 \sim \mathrm{Unif} \left( \frac{1}{n} \mathcal{S}^{n-1} \right)$. We compare $\|\nabla f(\boldsymbol{x}_k)\|$ obtained by different methods.