# OpenReview forum: "Greedy and Random Quasi-Newton Methods with Faster Explicit Superlinear Convergence"
_NeurIPS.cc/2021/Conference — NeurIPS 2021 Poster_

### Official Review · Reviewer_h3n6 · 2021-07-15

**Rating:** 6
**Confidence:** 4

**Summary:**

In this paper, the authors have applied the methods of greedy/random choice of directions that enable better Hessian approximation (proposed by Rodomanov and Nesterov in [18]) to the quasi-Newton optimization scheme. In this way, the authors have achieved a condition-number free super-linear convergence rate.

**Main Review:**

The paper is clearly written and the mathematics is clear. The only concern is that the improvment over Rodomanov and Nesterov in [18] might be incremental, so it would be a good idea if the authors can accompany the paper with a section indicating explicitly the improvements over Rodomanov and Nesterov in [18].

**Time Spent Reviewing:**

1 hour

---

> ### Author Response · Authors · 2021-08-09
> **Response to Reviewer h3n6**
>
> We thank the reviewer for review and support.
>
> Due to the page limit, we only list a brief comparison with [1] in our contribution summary and Table 1.
> We will include more content for comparison in the revised version.
>
> Our improvements over Rodomanov and Nesterov [1] mainly have two aspects.
> - The first aspect reflects directly on the condition-number dependence.
> The main superlinear convergence term in [1] is $\left(1-\frac{1}{n\kappa}\right)^{k(k-1)/2}$, but ours is condition-number-free: $\left(1-\frac{1}{n}\right)^{k(k-1)/2}$.
> - The second aspect is that we need fewer epochs ($k_0$) to reach the local superlinear convergence region.
> We need $k_0=O(\max \\{n, \kappa \\} \ln(n\kappa))$, but Rodomanov and Nesterov [1] needs $k_0=O(n\kappa\ln(n\kappa))$.
>
> Moreover, we solve two open problems mentioned in [1].
> - The first one is that Rodomanov and Nesterov [1] observed the random direction update also performs well, but they do not provide a theoretical guarantee.\
> The reviewer can refer to the last paragraph of Section 5.1 in [1]:
> ''We see that the randomized methods are slightly slower than the greedy ones. However, the difference is not really significant, and, what is especially interesting, the randomized methods do not loose superlinear convergence.''\
> We show that random direction updates could have a similar superlinear convergence rate with a theoretical guarantee.
> - The second one is that Rodomanov and Nesterov [1] observed that SR1 is faster than  BFGS, and DFP is the worst. \
> The reviewer can refer to the first paragraph on Page 22 (arxiv version) in [1]: ''Among quasi-Newton methods (both the standard and the greedy ones), SR1 is always better than BFGS, while DFP is significantly worst than the other two.'' \
> We show that (greedy and random) SR1 is faster than BFGS in the quadratic objective case (Theorem 3.2). And SR1 and BFGS could have condition-number-free superlinear convergence rates for strongly self-concordant objective, which in some sense is better than the rate of DFP obtained in [1] (see Table 1).
>
> [1] Anton Rodomanov and Yurii Nesterov. Greedy quasi-newton methods with explicit superlinear convergence. SIAM Journal on Optimization, 31(1):785–811, 2021. (https://arxiv.org/abs/2002.00657)

---

### Official Review · Reviewer_87Me · 2021-07-16

**Rating:** 7
**Confidence:** 4

**Summary:**

This  paper proposes a tighter local analysis of two popular variants of quasi-Newton methods: BFGS and SR1 with greedy or random choice of directions. It is heavily based on a recent work of Rodomanov & Nesterov, but the analysis is simpler and, most importantly, the guarantees are strictly better (the final bounds do not contain the conditional number $\kappa$).

I think it is really a good paper.


**Ethical Concerns:**

0

**Ethics Review Area:**

["I don’t know"]

**Limitations And Societal Impact:**

0

**Main Review:**

  Similarly to Rodomanov & Nesterov, the authors first study quadratic case (where the Hessian doesn't change from iteration to iteration) and then consider a general case with strongly self-concordant functions (where the Hessian changes in a controlled way). For SR1 update they use a slightly different (and simpler) measure function, which was key in establishing tighter bounds.



I checked mathematics only partially, mostly related to SR1 update. Although I still need some clarification on that, the major derivations seem correct.


__Proof of Theorem 4.3__:

1. line 451: why $M\lambda_k < 2$? We need it in order to apply Lemma B.2
2. Eq. (44): why do we write the last term in all inequalities as $\eta_k + 2Mr_k$  and not $\eta_k + 2M\lambda_k$ that we obtained in (41)? We need to use the latter anyway in the end of (44).
3. line 465: I find it a bit too quick. Also, we didn't use just (43), some derivations from (44) were also used.

__Proof of Corollary 4.4__:
1. Eq. after line 470: from that inequality we can only obtain the lower bound for $k_1$. Similarly for $k_2$ below.



__Minor comments__:
1. line 40: I think this work is not about closing the gap in the experiments, it is a theoretical work.
2. Algorithm 1, line 2: it is better to write "for $k = 0,\dots, n-1$"
3. I would add the updates for InvSR1 and InvBFGS to the main text.
4. line 171: there is no training process, it is just pure optimization.
5. Section 4: $\kappa$ has to be redefined here, before it was defined only for quadratics.
6. line 244: Is there any particular reason why we use the Frobenius norm for $X$ and not the spectral one (the smaller one)?
7. line 377: it is true, but for me it wasn't obvious.




The English of the paper might be improved, I list some errors below (maybe I didn't understand something).

1. Too many sentences start from "And", often there is no need for it.
2. line 44: "both the methods"
3. line 86: I am not sure what the authors want to say by "benign monotonicity"
4. line 103: "we would reuse"
5. line 111: "from the"
6. line 208: we do not verify theorems in optimization by numerical experiments.
7. line 232-233: "all obtain", "Theorem shows in Figure".
8. line 385 and in other places: not sure what "expectation view" means.
9. line 388: "are spherical symmetry"



**Time Spent Reviewing:**

0

---

> ### Author Response · Authors · 2021-08-09
> **Response to Reviewer 87Me**
>
> We thank the reviewer for your responsible review.
> Here are responses to the reviewer's comments.
>
> Answer for Proof of Theorem 4.3:
>
> 1. Line451: We are particularly grateful to the reviewer for discovering the missing detail.
> The condition $M\lambda_k<2$ builds on Theorem G.1. \
> Note that Theorem G.1 holds for Algorithm 8 (Scheme (4.17) of [1]) with arbitrary direction $u_k$ and quasi-Newton update in the Broyden family.
> And our Algorithm 4 is a special case of Algorithm 8. Thus Theorem G.1 still holds in our proof.
> Hence, when the initial condition $M\lambda_f(x_0) \leq  \frac{\ln\frac{3}{2}}{4\kappa}$, we could obtain the linear convergence of $\lambda_f(x_k)$, showing that $M \lambda_k \leq M \left(1-\frac{\mu}{2L}\right)^k \lambda_0 \leq M\lambda_0 \leq  \frac{\ln\frac{3}{2}}{4\kappa} < 2$ always holds.
> Our superlinear convergence builds on such facts since the initial condition $M\lambda_f(x_0)$ in Theorem 4.2 and Theorem 4.3 is stronger than Theorem G.1.
> Hence we could apply Lemma B.2 in our proof.\
> We will include more details to make the proof clear in a revised version.
>
> 2. Eq.(44): This is a typo, all $r_k$s in Eq.(44) should be $\lambda_k$s.
>
> 3. Line 465: We will include more details to clarify the use of previous equalities.
>
> Answer for Proof of Corollary 4.4:
>
> We apologize that the statement is incomplete. \
> Line 470: It should be: ''Denote $k_1\geq 0$, for which''  $\to$  ''Denote by $k_1\geq 0$ the number of the first iteration, for which''. And the other terms ($k_2$) should also be defined as above.
>
> $k_1$ describes the least epochs for satisfying the initial condition of Theorem 4.2 or Theorem 4.3 from the initial condition in Theorem G.1. $k_2$ describes the least epochs for making the superlinear convergence rate meaningful (e.g., smaller than 0.5).
>
> Answer for Minor comments:
>
> We thank the reviewer for pointing out several typos and improvements. We will handle them and polish the paper after revision.
>
> [1] Anton Rodomanov and Yurii Nesterov. Greedy quasi-newton methods with explicit superlinear convergence. SIAM Journal on Optimization, 31(1):785–811, 2021.

---

> > ### Comment · Reviewer_87Me · 2021-08-29
> > **Update**
> >
> > I am satisfied with the authors' response to both my and others reviews and will therefore keep my score.

---

### Official Review · Reviewer_pExt · 2021-07-16

**Rating:** 7
**Confidence:** 4

**Summary:**

This work considers the topic of the convergence of quasi-Newton and it builds on top of a series of recent results by Rodomanov and Nesterov. The main two contributions of this paper are: 1) improve the bounds for greedy quasi-Newton methods using a new potential function; 2) to propose a new modification of greedy strategy to obtain better results. The authors consider two settings: unconstrained quadratics and general strongly self-concordant functions (the same settings were considered in prior work). All results are local but this is probably inevitable as quasi-Newton methods approximate Newton's method, which itself is a local method.

I find the results to be very solid and promising. The topic of quasi-Newton methods might get a second breath due to the recent results and I would be glad to see this paper accepted. It does have a small number of typos but I think most of them can be fixed if the authors proofread their work (I hope the authors will do at least a partial pass during the discussion stage).

**Limitations And Societal Impact:**

Yes.

**Main Review:**

## Strength
1. This work has a very interesting theory and I particularly appreciate that the authors managed to use a somewhat simpler potential than the one employed by Rodomanov and Nesterov.
2. I also enjoyed the style of writing and the presentation of the obtained results.
3. I find the topic of quasi-Newton to be quite promising right now, and I hope that this work will inspire further research in this direction.
4. Quasi-Newton methods are widely used in practice but there are still many limitations (such as stochastic extensions of quasi-Newton methods), which helpfully can be addressed once we have a better understanding of their basic variants.


## Weaknesses
1. The paper has a number of typos and the use of articles is not great. I would also appreciate if the authors made some of the sentences shorter as it's hard to read them (for instance, the last sentence in the first paragraph).
2. The experiments are performed on logistic regression, which is often not a good example because it's well-conditioned. Please do similar experiments on the log-sum-exp objective, whose conditioning can be controlled using the smoothing parameter (by smoothing I mean approximation of the maximum function).

## Questions
In the proof of Corollary 4.4 (the part about SR1), the authors combine their guarantees with the guarantees of Rodomanov and Nesterov. However, when greedy strategies are used, I think this is not correct because Rodomanov and Nesterov use a different strategy for selecting u_k. Therefore, I think Corollary 4.4 is not precise in relation to SR1.

## Suggestions
1. As the results from Appendix C are not too long and they allow for a more efficient implementation that needs only O(n^2), I think they should be moved to the main part or at least explained briefly. I think this is particularly important because BFGS is sometimes more numerically stable than SR1.
2. Theorems 4.2 and 4.3 give slightly different results that suggest a better bound for BFGS than for SR1, for instance, in terms of the initial distance to the solution. I suggest adding this to Table 1.
3. Please update the numerical discussion to include log-sum-exp experiments.

## Typos
"quasi-Newton methods, that": no comma here
"Newton methods to certain approximation": "Newton methods with a certain approximation"
"worse long history behavior": "worse long-history behavior"
"Second, we extent": "Second, we extend"
"our analysis to practical scheme": "our analysis to a practical scheme"

**Time Spent Reviewing:**

3

---

> ### Author Response · Authors · 2021-08-09
> **Response to Reviewer pExt**
>
> We thank the reviewer for their review and support.
>
> Answer for Weaknesses:
>
> Thank you for your suggestions for paper writing.
> We would endeavor to shorten the sentences.
>
> After reading your review, we run the log-sum-exp objective with the same experimental settings as Section 5.1 of [1].
> We will add it in a later version.
>
> The experimental results are similar to the Logistic Regression.
> SR1-type methods are still much faster than BFGS-type methods.
> Moreover, we discover our RaBFGSv2 could be faster than RaBFGSv1 and GrBFGSv1 under really large condition numbers ($ 10^5 $).
> Such findings show that our method has less condition-number dependence, which partially supports our theory.
>
> Answer for Questions:
>
> We do not use the theorems obtained by the greedy strategy proposed by [1].
> The proof of Corollary 4.4 mainly uses the guarantees of Theorem 4.7 in [1] (i.e., Theorem G.1 in our paper).
>
> Theorem G.1 holds for Algorithm 8 with arbitrary direction $u_k$ and quasi-Newton update in the Broyden family.
> Since Algorithm 4 is still in the framework of Algorithm 8, we could combine the results of Theorem G.1 with our results.
>
> Rodomanov and Nesterov's [1] results under greedy strategy mainly have two optimization periods.
> The first period is proposed by Theorem G.1, which only has a linear convergence.
> The second period is proposed by their Theorem 4.9 in [1], which requires greedy strategies. And the second period indeed has a superlinear convergence but needs a stronger initial condition.
>
> In the proof of Corollary 4.4, we use Theorem G.1 as the first optimization period with a linear convergence rate.
> Note that Theorem G.1 has a weaker restriction on the initial point $x_0$ than Theorem 4.2 and Theorem 4.3.
> Hence, we need some epochs (that is $k_0$ shown in Table 1) to enter the second optimization period with a superlinear convergence rate proposed by Theorem 4.2 and Theorem 4.3.
>
> Therefore, we only use the first-period results of [1] instead of the second-period results produced by greedy strategies in [1].
>
> We thank you for pointing out several Typos and providing helpful suggestions. We will revise our paper in a later version accordingly.
>
> [1] Anton Rodomanov and Yurii Nesterov. Greedy quasi-newton methods with explicit superlinear convergence. SIAM Journal on Optimization, 31(1):785–811, 2021.

---

> > ### Comment · Reviewer_pExt · 2021-08-09
> > **Thank you for the clarification and good work!**
> >
> > Thank you for the clarification regarding the proof. I missed the part from [1] that the result applies to any selection of u_k and thought that it only works for their greedy selection. With this clarification, I agree that the proof of Corollary 4.4 is correct.
> >
> > Please do take extra care when proofreading the paper, it deserves some polishing.

---

### Official Review · Reviewer_8zir · 2021-07-20

**Rating:** 6
**Confidence:** 2

**Summary:**

This paper introduces new non-asymptotic analysis for quasi-Newton methods, extending recent results on DFP to BFGS and SR1. This is relevant since these two methods produce better Hessian approximations so can converge faster. The authors show that the same rates of convergence can be achieved using either greedy or random updates, which is relevant because greedy updates do not always come for free.

**Limitations And Societal Impact:**

I see no additional societal impact to the existing literature on the topic

**Main Review:**

The paper introduces interesting new analysis which advances the understanding of quasi-Newton methods. Its outline is relatively straightforward. The theoretical results are written in a way that the reader can see its important parts, rather than being overcome with complex constants (which sadly happens quite often in the related literature). The theoretical results are convincing.

On the other hands, a few concerns appear.
First, the clarity of the paper could be improved upon. There are three steps: matrix approximation, then minimisation of a very simple quadratic objective (with fixed Hessian), and finally optimisation of a more. relevant ML objective. Making this gradation more explicit would help the non-expert reader figure out what exactly is being optimised at any given moment, which can prove a bit tricky for now.

Second, a lot of energy and space is devoted to developing theory on very simple problems on which the algorithms will not be applied (if there's a closed form solution, why bother optimising?). It appears at the very end of the paper that on relevant tasks, the new method does not perform better than previous work, which makes the whole endeavour somewhat less relevant.

All told, while this paper is not groundbreaking from a practical standpoint it presents convincing new theoretical analysis and advances our understanding of relevant methods.


Details:
- in the notations using x as the argument and index for the norm definition is confusing.
- In theorem 2.3, the inequality does not seem to imply super but rather sub-linear convergence, unless the G_0 is replaced by G_{k-1}?

**Time Spent Reviewing:**

2.5

---

> ### Author Response · Authors · 2021-08-09
> **Response to Reviewer 8zir**
>
> Thank you for your review. We will improve the clarity in a revised version based on your suggestions.
>
> We consider theoretical results should build on the simple case and have a strong intuition to make readers follow up.
> Therefore, we begin with matrix approximation, then quadratic objective (with fixed Hessians), finally to a relevant ML objective (with unfixed Hessians).
>
> Moreover, general relevant tasks should also include the simple quadratic objective. Since around the optimal point, we could view the objective as a quadratic objective by Taylor expansion.
> Hence, the results under the quadratic objective could indeed show some intuition of such superlinear convergence, though there's a closed-form solution in the quadratic case.
> Furthermore, even though we have a closed-form solution in the quadratic case, we need to compute the inverse matrix, which needs $O(n^3)$ complexity.
> Therefore, there are many numerical optimization methods even for the quadratic case, rather than applying the analytic solution.
>
> Finally, we have verified the benefit of our proposed method in the quadratic case (Figure 1c).
> Hence we consider our scaled direction is more suitable for a constant Hessian matrix which appears in the late optimization period that the Hessian maintains approximately fixed.
> As Figure 2 depicted, our RaBFGSv2 decreases faster than RaBFGSv1 in the last few epochs.
>
> By the way, as proposed by other reviewers, we include the log-sum-exp objective experiments as Section 5.1 of [1] did.
> We discover that for a large condition-number ($10^5$), our RaBFGSv2 could be faster than RaBFGSv1 and GrBFGSv1.
> We also try large-condition-number experiments in Logistic Regression, finding the same experimental results.
> Therefore, we believe our proposed methods have less condition-number dependence as our theory shows.
>
> Answer for Details:
>
> - The notations $|| \cdot ||_{x}$ is inherited from [1], we also consider it is confusing.
>
> - In theorem 2.3, we consider the inequality implies superlinear convergence.
> We plot such upper bound with the 'theory' legend (green line) in Figure 1a (note that the vertical coordinate is already a log scale).
> We can observe the green curve is faster than a straight line, showing a superlinear convergence.
>
> [1] Anton Rodomanov and Yurii Nesterov. Greedy quasi-newton methods with explicit superlinear
> convergence. SIAM Journal on Optimization, 31(1):785–811, 2021.

---

> > ### Comment · Reviewer_8zir · 2021-08-22
> > **Thanks for your answer**
> >
> > The answer clears up a few concerns I had. I still encourage the authors to contextualise the obtained results in terms of machine learning application, as they have done in the comment, early in the paper, which would help for less expert readers.
> > The new results with less well-conditioned objective are a good addition to the paper.
> > I agree with the other reviewers that given a nice polishing the paper can be published at NeurIPS this year.

---

### Official Review · Reviewer_y3vR · 2021-08-03

**Rating:** 6
**Confidence:** 4

**Summary:**

This paper presents two modified greedy quasi-Newton updates, originally given by Rodomanov and Nesterov [18]. They first consider using these updates to approximate a positive definite Hessian. In particular, for the greedy SR1 update, they introduce a different progress measure, which is maximized over the coordinate bases to yield the vector used in the update. This new measure allows them to obtain a superlinear convergence rate for the Hessian approxiamtion error, where the rate is independent of the condition number $\kappa$. For the greedy BFGS update, they use the same progress measure as in Rodomanov and Nesterov [18], but preconditions the corresponding greedily selected vector with the current Hessian approximation. This yields a linear convergence rate for the Hessian approximation, although their greedy selection for BFGS is rather impractical. In addition to the modified greedy selection strategies, their analyses and rates also apply to a random selection from a spherically symmetric distribution.

Similar to Rodomanov and Nesterov [18], they then apply these results to analyzing the convergence rate of minimizing strongly-convex quadratics, as well as strongly self-concordant objectives. They obtain similar explicit local rates as the previous work, but without the condition number (although the initial, non-local phase still converges linearly with a dependence on $\kappa$).

Experiments on evaluating the newly introduced Hessian approximations show that their methods are quite effective. On a strongly-convex quadratic minimization problem the conclusions are similar. For minimizing a strongly-convex logistic regression problem, the randomized BFGS (with the preconditioned vector) does not perform better than the version from Rodomanov and Nesterov [18], which does not contain the preconditioning. This is true for the two logistic regression problems with different scales of condition numbers.


**Limitations And Societal Impact:**

Yes, the authors have adequately addressed the limitations and potential negative societal impact of their work.

**Main Review:**

**Strengths**
This work is strong in the sense that they are able to significantly improve existing rates with very simple modifications to an existing algorithm, especially when the improved (local) rates do not depend on the condition number. These results are very interesting and further establish the potential of greedy/random quasi-Newton methods. The paper is easy to read as it mainly follows the same structure as Rodomanov and Nesterov [18], with key differences emphasized appropriately.

**Weaknesses**
One of the drawbacks in the proposed greedy updates is that we now have two different greedy selection rules for SR1 and BFGS, respectively. This is not a main issue in terms of implementation ease, as the two Hessian approximation updates themselves are already different. However, this distinction leads to separate analyses for SR1 and BFGS, which is not as satisfying as the analyses provided by Rodomanov and Nesterov [18], where the three main quasi-Newton updates (SR1, BFGS, DFP) are all covered. Another drawback of this distinction is that, the proposed greedy BFGS is rather impractical, as the authors mentioned that the cost to obtain the vector is $O(n^3)$.  Furthermore, this work does not seem to cover the DFP update, which leaves me wondering whether the new progress measure introduced for SR1 or the preconditioning of the greedy vector in the BFGS case are mainly ad hoc, and whether there is a way to obtain condition number free rates for the general Broyden family.

=== Update ===
I no longer view this as a weakness. Unified analyses would be nice if they also provide tight rates, but it seems to be highly non-trivial for the Broyden family.

In terms of the related work, the authors should discuss the work of [Jin and Mokhtari, 2020], in which non-asymptotic superlinear convergence rate of the original Broyden family are also proved. In their results, the rates are independent of the problem dimension $n$. In addition, they do not use the greedy/random basis selection, nor do they require access to the diagonal of the true Hessian.

For the first set of experiments (Figure 1), it would be helpful to see a comparison between problems with varying condition numbers, and compare whether the local convergence of the proposed updates are indeed condition-number independent in practice.

**Reference mentioned**
[Jin and Mokhtari, 2020] *Non-asymptotic superlinear convergence of standard quasi-Newton methods*


**Minor concerns**
- B.3 Proof of Theorem 3.2
	- This proof invokes Lemma 3.1 (Lemma 3.2 from Rodomanov and Nesterov [18]), while assuming it holds with an expectation. I believe it indeed holds with an application of the operator convexity property of the matrix inverse, but it would help clarify if the authors can include the proof of this lemma under expectation.
- L102: The authors claimed that the progress measure $\sigma_\mathbf{A}(G)=\text{tr}(A^{-1}(G-A))$ might be improper for analyzing the SR1 update. Could you please elaborate why this might be true? Is it because SR1 is a rather good approximation of the exact Hessian, and thus preconditioning the progress measure with $A^{-1}$ could de-emphasize the proximity between $G$ and $A$? And hence the modification to $\tau_\mathbf{A}(G)=\text{tr}(G-A)$ ?

**Time Spent Reviewing:**

16

---

> ### Author Response · Authors · 2021-08-09
> **Response to Reviewer y3vR**
>
> Thank you for giving us constructive feedback. We list our answers to your comments/questions below.
>
> Answer for Weaknesses:
>
> 1. We do not consider the distinction analyses for SR1 and BFGS is not satisfying.
> As we mentioned in Line29-32, Rodomanov and Nesterov [1] reduced all possible Broyden
> family to the DFP update based on the monotonicity property.
> However, the SR1 and BFGS update is faster than the DFP update in practice.
> Moreover, Rodomanov and Nesterov's [1] experiments also confirm this phenomenon (see Section 5 of [1] for detail).
> Therefore, covering the three main quasi-Newton updates (SR1, BFGS, DFP) with the same superlinear convergence rate seems unreasonable.
> Hence, we expect faster superlinear convergence rates for SR1 and BFGS compared to DFP.
> Such consideration motivates us to find out the indeed faster convergence applied by SR1 and BFGS.
>
> 2. We recognize the proposed greedy BFGS is rather impractical which is mentioned in Remark 2.6.
> However, our proposed random BFGS is still practical with complexity $O(n^2)$ in each iteration and available in experiments.
>
> 3. We thank the reviewer for pointing out the reference [2] and will include it in a revised version. However, the results of [2] need an extra initial Hessian approximation condition (e.g. Eq. (59) in Theorem 4.5 of [2]).
> Based on this extra initial Hessian approximation condition, Jin and Mokhtari [2] could obtain a strong dimension-free superlinear convergence.
> Yet we consider a common initial Hessian approximation matrix ($L I_n$).
> Therefore our results are not covered by Jin and Mokhtari [2].
> However, we could not obtain dimension-free results since we do not have such an initial Hessian approximation condition. \
> By the way, the dimension-free superlinear convergence rate sounds interesting.
> We will take advantage of the excellent technique of [2] to strengthen our results in future work.
>
> 4. Giving the comparison between problems with varying condition numbers is a helpful suggestion.
> After reading your suggestions, we have run such experiments and will add them in a revised version. \
> Here, we briefly describe our discovery:
> Overall, our RaBFGSv2 (scaled random direction) matches our theory among various condition numbers, like Figure 1b shows in our current version.
> However, the RaBFGSv1 (original random direction) converges much slower than RaBFGSv2, particularly in large condition numbers.
> These findings support our condition-number-free convergence rate.
>
> Answer for Minor concerns:
>
> 1. We apologize that the original proof of Theorem 3.2 was not clear enough.
> We have reorganized the proof of Theorem 3.2 to help clarify.
> Thanks again for your responsible review.
>
> 2. Here 'improper' only means that we do not know how to obtain similar results for $\sigma_A(G)$.
> We have roughly the same understanding of not using $\sigma_A(G)$.
> The SR1 update is a rather good approximation of the exact Hessian, and thus preconditioning the progress measure with $A^{-1}$ could de-emphasize the proximity between $G$ and $A$. \
> Another insight to judge $\sigma_A(G)$ is improper comes from experiments.
> We also run the same experiments as Figure 1a in our paper under measure $\sigma_A(G)$.
> We discover the unstable behavior of such measure, leading us to select another measure.
>
> [1] Anton Rodomanov and Yurii Nesterov. Greedy quasi-newton methods with explicit superlinear convergence. SIAM Journal on Optimization, 31(1):785–811, 2021.
>
> [2] Jin and Mokhtari, 2020. Non-asymptotic superlinear convergence of standard quasi-Newton methods.

---

> > ### Comment · Reviewer_y3vR · 2021-08-12
> > **Thank you for addressing my comments.**
> >
> > As mentioned in the other thread, I now agree with the authors and the reviewers that the separate analyses for SR1 and BFGS should not be viewed as a weakness, but perhaps unavoidable to obtain a condition-number independent rate. I am also satisfied with the authors' response to my other comments, and therefore I will update my score accordingly.

---

> ### Comment · Reviewer_pExt · 2021-08-09
> **Regarding non-unified nature of the results**
>
> I mentioned that the reviewer points out the different analyses of SR1 and BFGS as a weakness. I hope it's ok if I say that this shouldn't be viewed as a weakness because SR1, BFGS and DFP have quite different numerical convergence. I think it's rather a weakness of the paper by Rodomanov and Nesterov that they use a unified analysis because it makes some special cases not tight.

---

> > ### Comment · Reviewer_y3vR · 2021-08-12
> > **Agreed**
> >
> > I think you have a point here that adopting a unified analysis could be making the rates loose, and thus the separate analyses for SR1 and BFGS should not be viewed as a weakness. I will update my review accordingly. Thank you for pointing this out!

---

### Decision · Program_Chairs · 2021-09-27

**Decision:**

Accept (Poster)

**Comment:**

All reviewers recommended acceptance, and this represents as a non-trivial improvement over the existing results. Please make the changes discussed in the reviews/responses, such as the addition of the log-sum-exp experiment.